# Laser-Induced Graphene on Biocompatible PDMS/PEG Composites for Limb Motion Sensing

**DOI:** 10.3390/s25175238

**Published:** 2025-08-22

**Authors:** Anđela Gavran, Marija V. Pergal, Teodora Vićentić, Milena Rašljić Rafajilović, Igor A. Pašti, Marko V. Bošković, Marko Spasenović

**Affiliations:** 1Center for Microelectronic Technologies, Institute of Chemistry, Technology and Metallurgy—National Institute of the Republic of Serbia, University of Belgrade, 11000 Belgrade, Serbia; marija.pergal@ihtm.bg.ac.rs (M.V.P.); teodora.vicentic@ihtm.bg.ac.rs (T.V.); milena.rasljic@ihtm.bg.ac.rs (M.R.R.); marko.boskovic@ihtm.bg.ac.rs (M.V.B.); 2Faculty of Physical Chemistry, University of Belgrade, 11000 Belgrade, Serbia; igor@ffh.bg.ac.rs; 3Serbian Academy of Sciences and Arts, 11000 Belgrade, Serbia

**Keywords:** laser-induced graphene, poly(dimethylsiloxane), poly(ethylene glycol), physicochemical properties, sensors

## Abstract

The advancement of laser-induced graphene (LIG) has significantly enhanced the development of wearable and flexible electronic devices. Due to its exceptional physical, chemical, and electronic properties, LIG has emerged as a highly effective active material for wearable sensors. However, despite the wide range of materials suitable as precursors for LIG, the scarcity of stretchable and biocompatible polymers amenable to laser graphenization has remained a persistent challenge. In this study, laser-induced graphene (LIG) was fabricated directly on biocompatible and flexible cross-linked PDMS/PEG (with M_n_ (PEG) = 400 g/mol) composites for the first time, enabling their application in wearable sensors. The addition of PEG compensates for the low carbon content in PDMS, enabling efficient laser graphenization. Laser parameters were systematically optimized to achieve high-quality graphene, and a comprehensive characterization with varying PEG content (10–40 wt.%) was conducted using multiple analytical techniques. Tensile tests revealed that incorporating PEG significantly enhanced elongation at break, reaching 237% for PDMS/40 wt.% PEG while reducing Young’s modulus to 0.25 MPa, highlighting the excellent flexibility of the substrate material. Surface analysis using X-ray photoelectron spectroscopy (XPS), scanning electron microscopy (SEM), and Raman spectroscopy demonstrated the formation of high-quality few-layer graphene with the fewest defects in PDMS/40 wt.% PEG composites. Nevertheless, the adhesion of electrical contacts to LIG that was directly induced on PDMS/PEG proved to be challenging. To overcome this challenge, we produced devices by means of laser induction on polyimide and transfer to PDMS/PEG. We demonstrate the practical utility of such devices by applying them to monitor limb motion in real time. The sensor showed a stable and repeatable piezoresistive response under multiple bending cycles. These results provide valuable insights into the fabrication of biocompatible LIG-based flexible sensors, paving the way for their broader implementation in medical and sports technologies.

## 1. Introduction

In recent years, significant progress has been made in the research and development of wearable sensors, driven by the increasing demand for stretchable and flexible wearable devices as an alternative to traditional rigid systems. These advances have led to the development of wearable skin patches that monitor various physiological parameters such as heart rate, respiration, blood pressure, speech and limb movements for applications in medicine and sports [1,2,3]. To meet the requirements for sensors worn on the skin, the materials must be thin and flexible and have desirable electrical properties, such as high conductivity [4]. One material that has properties that fulfill these requirements is laser-induced graphene (LIG). The laser induction process gained a lot of attention in 2014 when polyimide (PI) was directly converted into porous graphene under CO_2_ laser irradiation [5]. Laser induction was presented as a highly controlled technique for the conversion of carbon-rich parts of polymers into graphene. The result is a graphene/polymer heterostructure in which graphene can be used as a sensor and the polymer serves as a flexible, mechanically stable substrate.

The optimization of laser parameters such as scanning speed, power and resolution can lead to the induction of graphene with micro- and nanoscale structures as well as tunable conductivity, surface morphology and chemical composition. Although the electrical properties of LIG are inferior to those of graphene produced by other methods such as chemical vapor deposition and mechanical exfoliation, the simplicity and low cost of fabrication make laser induction a highly attractive approach [6]. The remarkable properties of LIG, such as its large surface area (428 m^2^ g^−1^), good flexibility, piezoresistive properties, high thermal stability (>900 °C), high mechanical stability, good electrical conductivity (5–25 S cm^−1^), and ease of fabrication, make it an ideal candidate for wearable sensors [5,6,7,8,9,10,11,12]. Moreover, the properties of this material can be tuned by doping with heteroatoms or nanoparticles to improve the electrocatalytic activity, electrochemical performance or surface wettability of LIG [13,14,15]. The structure and physicochemical properties of LIG have become the focus of researchers who want to develop flexible sensors.

Wang et al. demonstrated LIG-based sensors on PI/PDMS for monitoring electrophysiological activities, detecting limb movements, and as a tool for remote control of an actuator [1]. LIG-based sensors fabricated on PI/PDMS composite substrates exhibit improved stretchability and can withstand mechanical stresses of over 15%, which is essential for wearable electronics applications [1]. The tunable conductivity and surface morphology of LIG enable customization of sensor performance, resulting in high sensitivity and fast response times [1,16]. In addition, the porous and flexible nature of LIG helps to maintain structural integrity across repeated deformation cycles, which is critical for reliable finger movement detection [17,18]. These properties make LIG-based strain sensors promising candidates for applications in health monitoring, robotics and human–machine interfaces [1,19,20].

Commercially available PI films are the most common substrate for the production of high-quality LIG due to their good mechanical stability and high thermal resistance (up to 400 °C) [6,21,22]. PI is widely used in biomedical applications due to its thermal stability, mechanical properties and chemical resistance, especially in implantable microelectrodes, flexible neural interfaces and microfluidic devices. Polyimide, although successfully used in biomedical devices, often requires additional surface treatments or coatings to improve cell compatibility or reduce inflammatory reactions, and the elasticity of PI is low [23,24,25,26,27]. Therefore, LIG is usually transferred from PI to other polymers to improve flexibility and extensibility [3,28]. Polymers with similar structures to PI, e.g., polyetherimide or polysulfone, have been reported to be suitable for LIG [5,12,14,29,30]. LIG have also been prepared from other carbonaceous materials such as wood, cloth, paper, food and alginate [31,32,33,34]. However, the fabrication of LIG/polymer heterostructures that are biocompatible, conductive and stretchable remains a major challenge [34,35].

Poly(dimethylsiloxane) (PDMS) is a biocompatible and stretchable silicon-based polymer that is widely used as an elastomer in wearable applications due to its excellent mechanical properties, optical transparency, tunable surface chemistry, low water permeability, excellent thermal, oxidative and UV stability, and high gas permeability [36]. Due to its low cost, biocompatibility, flexibility and unique surface properties, it is used in many different applications, especially in microfluidics, flexible electronics and tissue engineering [37,38,39,40,41,42,43,44,45,46]. Some of these properties, such as low surface energy, low chemical reactivity and hydrophobicity, enhance the biocompatibility of PDMS [47]. The flexibility of PDMS results from the structural properties of the Si-O bonds [47]. Since PDMS consists mainly of silicon and oxygen atoms in the main chain, with side methyl groups attached to the silicon atom, it is not suitable as a carbon precursor for laser-induced graphene. PDMS tends to decompose upon laser irradiation (>900 °C) instead of forming graphene [48]. Therefore, researchers are looking for ways to improve the properties of PDMS by introducing another polymer or carbonaceous precursor into the PDMS matrix, e.g., poly(ether ether ketone) (PEEK), triethylene glycol (TEG), polyimide particles and Triton X-100 [13,36,49,50]. By introducing TEG or PEEK as a carbon source into the PDMS matrix, one can improve the graphenization and reduce the sheet resistance of the LIG/PDMS-based composite [13]. Parmeggiani et al. fabricated PI/PDMS substrates by mixing PI powder with PDMS and then irradiating with a CO_2_ laser to produce LIG for flexible strain sensors [49]. The PI particles are graphenized, and in these composites the conductivity remains limited by the connection between the individual PI particles.

One polymer that can be an excellent carbon source when incorporated into a PDMS matrix is poly(ethylene glycol) (PEG). PEG is a versatile synthetic polymer consisting of repeating ethylene oxide units (-CH_2_CH_2_O-) and has unique properties such as water solubility, non-toxicity, good biocompatibility, hydrophilicity and chemical stability, which makes PDMS/PEG composites particularly attractive for wearable applications with direct skin contact [51,52,53]. Thanks to its unique properties, it can be used in the pharmaceutical industry, biotechnology and medicine [51,53,54,55]. For example, a PDMS-PEG block copolymer with drug pretreatment has been used to reduce drug absorption by 91.6% and enable accurate drug screening [56]. Furthermore, the combination of PDMS and PEG has shown promise for applications such as CO_2_ gas separation membranes as it utilizes the complementary properties of these polymers [57]. However, to date, there are no studies reporting the fabrication of cross-linked PDMS/PEG substrates for laser-induced graphenization. Recent reports by Pinheiro et al. and Claro et al. have emphasized the importance of exploring both the physical mechanisms and sustainable precursors for LIG formation [58,59].

In this context, our study represents an advance by demonstrating the direct formation of laser-induced graphene on novel biocompatible PDMS/PEG substrates with optimized processing parameters and application-ready performance. To our knowledge, this is the first report showing direct laser induction of graphene on cross-linked PDMS/PEG substrates, expanding the material landscape for biocompatible LIG-based flexible electronics. The key novelty and innovation of our research lies in the direct formation of LIG on cross-linked PDMS/PEG composites and the application of LIG transferred from PI to PDMS/PEG for limb motion monitoring. We demonstrate laser induction of graphene on novel biocompatible PDMS/PEG substrates with different PEG content. We optimize the laser parameters for different PEG contents in the PDMS matrix. We perform physicochemical characterization of the PDMS/PEG composites and the directly graphenized PDMS/PEG materials, including electrical resistivity measurements, Raman spectroscopy, Fourier transform infrared spectroscopy (FTIR), scanning electron microscopy (SEM) with energy dispersive X-ray analysis (EDX), X-ray diffraction (XRD), swelling behavior, mechanical characterization, environmental stability (humidity and temperature), water contact angle measurements, water absorption, X-ray photoelectron spectroscopy (XPS), transmission electron microscopy (TEM) and thermogravimetric analysis (TGA). We find the optimal PEG content and laser parameters for effective graphene induction. Furthermore, we demonstrate the application in limb motion sensing by transferring LIG from PI to PDMS/PEG to establish stable electrical contacts to the devices, which is an unresolved challenge in direct induction on PDMS/PEG. This work highlights the potential of biocompatible, flexible and efficient LIG-based sensors for limb motion monitoring. Although the adhesion of electrical contacts to directly induced LIG on PDMS/PEG remains a challenge, we provide clear evidence that the LIG formed is conductive, structurally well-defined, and capable of sensing under mechanical deformation. A representative sensor fabricated with directly induced LIG showed piezoresistive response under strain, confirming the intrinsic sensing capability of the material. However, to evaluate device stability under repeated motion cycles, we employed transferred LIG from PI due to its better contact adhesion, while retaining the same PDMS/PEG substrate.

## 2. Experimental Section

### 2.1. Materials

Poly(dimethylsiloxane) (PDMS) prepolymer and curing agent (Sylgard 184, Dow Corning, Miami, FL, USA), along with poly(ethylene glycol) (PEG) (*M*_n_ = 400 g/mol, Sigma Aldrich, Darmstadt, Germany), were used to synthesize the precursor. Toluene (Lach:ner, Neratovice, Czech Republic) was employed to investigate swelling behavior.

Polyimide (DuPont Kapton^®^, Wilmington, DE, USA) with a thickness of 75 µm was used for fabricating LIG for transfer onto PDMS/PEG in the final sensor application test. Transfer from PI to PDMS/PEG was necessary for the application to limb motion sensing due to the unsolved issue of electrical contact adhesion to LIG directly induced on PDMS/PEG.

### 2.2. Synthesis of PDMS/PEG Composites

The PDMS prepolymer (α,ω-divinyl poly(dimethylsiloxane)) (PDMS base) was mixed with the curing agent (poly(methyl-hydrogensiloxane)) (PMHS) in a precise weight ratio of 10:1 (Figure 1a). Following this, different mass percentages of PEG (10, 20, 30, and 40 wt.%) were added to the mixture (Figure 1b). The mixture was stirred for 15 min to ensure homogeneity. The reaction mixture was then poured into a Petri dish and degassed in a vacuum-drying chamber for 30 min. Subsequently, the reaction mixture was transferred to an oven and left for cross-linking at 100 °C for 3 h. To complete curing, the samples were subjected to an additional vacuum drying step at 50 °C for 5 h the following day. For comparison, a pure PDMS network was synthesized under the same conditions.

### 2.3. Laser-Induced Graphene Production and Laser Parameter Optimization Strategy

The CO_2_ laser used to prepare LIG was a DBK FL-350 (DBK, Radije ob Dravi, Slovenia) with a wavelength of 10.6 μm. The maximum power output of the laser is 60 W, and the resolution was fixed at 1200 DPI. For each PEG concentration (10, 20, 30, 40 wt.%), at least three separate samples were processed, with LIG induced in a square geometry (3 × 3 mm).

Starting with a range of laser power (9–10.8 W), speed (35–55 mm s^−1^), and fixed resolution (1200 DPI), we conducted iterative trials on PDMS/PEG composites with 10–40 wt.% PEG content. For each set of conditions, we evaluated (i) visual appearance (shade of black and continuity of LIG) and (ii) electrical resistance (using a multimeter).

### 2.4. LIG Characterization

Electrical measurements were conducted using a multimeter. The probes were spaced 3 mm apart.

The Raman spectra of the materials were recorded using a DXR Raman microscope (Thermo Fisher Scientific, Waltham, MA, USA) with a laser wavelength of *λ_l_* = 532 nm and a power of 2 mW, focused onto a 2.1 μm spot on the surface. Three measurements at different positions on each sample were performed (10 exposures, 10 s each per position). The crystalline size along the a-axis (*L_a_*) was calculated using the Tuinstra and Koenig equation as follows [60,61,62,63]:(1)$$L_{a}=(2.4\cdot 10^{-10})\cdot \lambda_{l}^{4}\cdot ({\frac{I_{D}}{I_{G}})}^{-1}$$

SEM combined with EDX (Phenom, Thermo Fisher Scientific, Waltham, MA, USA) was employed to analyze material morphology and chemical composition.

The FTIR spectra of PDMS/PEG and powdered LIG, obtained by scraping from PDMS/PEG composites, were recorded in transmission mode with an FTIR spectrophotometer (Thermo Fisher Scientific, Waltham, MA, USA) on a KBr substrate. Pure PDMS was analyzed with a Nicolet 6700 spectrometer (Thermo Scientific, Waltham, MA, USA) using attenuated total reflection (ATR) mode equipped with a diamond crystal, and spectra were corrected with ATR correction. All spectra were collected in the range from 4000 to 400 cm^−1^, with a resolution of 4 cm^−1^ and 32 scans for each sample.

XRD was conducted using a Ultima IV diffractometer (Rigaku, Tokyo, Japan). The X-ray beam was nickel-filtered CuK_α1_ radiation (*λ* = 0.154 nm) operating at 40 kV and 40 mA. XRD data were collected from 3 to 50° (2θ) at a scanning rate of 2° min^−1^, with a step size of 0.02°. To provide high-intensity, high-resolution measurements, parallel beam geometry and the D/teX Ultra, a high-speed one-dimensional X-ray detector (Rigaku, Tokyo, Japan), were used.

Swelling measurements were performed at room temperature by immersing the materials in toluene for 48 h. The mass of a swollen sample and material mass before swelling were measured and then used to calculate the swelling degree, *q_e_*, in the following manner:(2)$$q_{e}=\frac{w\;-\;w_{0}}{w_{0}}$$

Using these results of swelling measurements and the Flory-Rehner equation for an equilibrium swollen network, the cross-linking density was calculated as follows [64]:(3)$$v=-\frac{lnln\;\left(1\;-\;V_{PDMS/PEG}\right)\;\;+\;V_{PDMS/PEG\;}\;+\;\chi V_{PDMS/PEG}^{2}}{V_{s}(V_{PDMS/PEG}^{\frac{1}{3}}-\;\frac{V_{PDMS/PEG}}{2})}$$
where *ρ* = 0.45 is the polymer-solvent interaction parameter, *V_s_* is the molar volume of toluene (106.288 cm^3^ mol^−1^), and *V_PDMS/PEG_* represents the volume fraction of the sample in the swollen sample. Knowing the density of PDMS/PEG (*ρ_PDMS/PEG_*), the density of toluene (*ρ_s_* = 0.867 g cm^−3^) and masses of the swollen sample (*w*) and mass of a sample measured after 48 h in a vacuum dryer when the mass reached constant value (*w_g_*), the value of the volume fraction *V_PDMS/PEG_* could be calculated following:(4)$$V_{PDMS/PEG}=\frac{1}{1\;+\;\frac{\rho_{PDMS/PEG}}{\rho_{s}}\;\cdot \;\frac{w\;-\;w_{g}}{w_{g}}}$$

The average molecular mass of the polymer chain between cross-links was calculated according to Equation (5):(5)$$M_{c}=\frac{\rho_{PDMS/PEG}}{v}$$

Mechanical properties were studied on a Universal Testing Machine (Shimadzu AGS-X, Kyoto, Japan) using an elongation speed of 10 mm min^−1^ for testing tensile strength and elongation at break and a speed of 5 mm min^−1^ for measurements of Young’s modulus.

The gauge factor (GF) was evaluated to estimate the sensor piezoresistivity with the following equation:(6)$$GF=\frac{R_{\varepsilon }-R_{0}}{R_{0}}\cdot \frac{1}{(L_{\varepsilon }-L_{0})/L_{0}}$$
where *R_0_* is the sensor resistance without deformation, *R_ε_* is the sensor resistance after applying a strain *ε*, *L*_0_ is the initial length of the sensor, and *L_ε_* is the length of the sensor after applying strain *ε*. Measurements were performed with the 34461A (Keysight, Santa Rosa, CA, USA), which was interfaced with a desktop computer. Sensor resistance was measured while applying strain in 2% increments up to a maximum of 10%.

All electrical and mechanical tests were performed at ambient temperature (22–24 °C) and relative humidity of 40–50%. Hydrophobicity was investigated through surface tension measurements with a contact angle goniometer (Ossila, L2004A1, Sheffield, UK) using the sessile drop method to determine the water contact angle. Five measurements were performed at room temperature using distilled water.

Water absorption (WA) was investigated by immersing materials in distilled water at room temperature for 48 h. After 48 h, the materials were removed and blotted with filter paper to remove excess water. The mass percentage of water absorption was calculated with Equation (7):(7)$$Water\;absorption=\;\frac{w_{w}-w_{w0}}{w_{w0}}\;\cdot 100$$
where *w_w_* is the weight of the fully hydrated material, and *w_w_*_0_ is the mass of the dried material.

XPS was performed with an XPS system (SPECS Systems, Berlin, Germany) equipped with an XP50M X-ray source, a Focus 500 X-ray monochromator, and a PHOIBOS 100/150 analyzer (SPECS Systems, Berlin, Germany). Measurements utilized an AlK_α_ source (1486.74 eV) at 12.5 kV and 16 mA. Survey spectra (1000–0 eV binding energy) were recorded with a constant pass energy of 40 eV, a step size of 0.5 eV, and a dwell time of 0.2 s in the FAT mode. Detailed spectra for C 1s, O 1s, and Si 2p peaks were obtained using a constant pass energy of 20 eV, a step size of 0.1 eV, and a dwell time of 2 s in the FAT mode. During measurements, the pressure in the chamber was 5 × 10^−8^ mbar. All the peak positions were referenced to C 1s at 284.8 eV.

LIG samples for TEM were prepared by scraping LIG off the substrate into a powder, dispersing it in 96% ethanol with the assistance of ultrasound, depositing the suspension onto a carbon-coated copper grid, and allowing the grid to dry at ambient temperature. TEM analysis of prepared LIG samples was performed with an FEI Talos F200X microscope (Thermo Fisher Scientific, Waltham, MA, USA) operating at 200 kV.

The change in electrical resistance was monitored at four different relative humidity levels (40, 60, 80 and 90%) at 25 °C and different temperatures (20, 30, 40 and 50 °C) with 40% humidity to investigate the influence of relative humidity (RH) and temperature on LIG. LIG/PDMS with different PEG content (20–40 wt.%) was fixed to the glass substrate and placed in the humidity test chamber (Environmental chamber—TH-50B (LIB, Xi’an, China)) (Appendix A) [65]. Composites with attached copper contacts were connected to the Keithley 2450 (Keithley, Solon, OH, USA) for resistance monitoring at given humidity levels. Stabilization for each set of RH took 20 min at room temperature (25 °C).

Thermogravimetric analysis (TGA) was carried out on a TGA Q500 thermogravimetric analyzer (TA Instruments, New Castle, DE, USA) under a nitrogen flow (purity stream of 99.999%) of 60 mL min^−1^ and a flow of 60 mL min^−1^ in the temperature range from 25 to 700 °C at a heating rate of 10 °C min^−1^.

### 2.5. LIG-Based Limb Motion Sensor Fabrication

The composite based on PDMS with 40 wt.% PEG was selected as the optimal substrate for LIG-based limb motion sensors. Two PDMS/PEG composites were synthesized: the first one contained 5 g of PDMS, 0.5 g of curing agent, and 2.2 g of PEG, and the second one contained 3 g of PDMS, 0.3 g of curing agent, and 1.32 g of PEG. The first sample had a thickness of 1.5 mm, and the second had a thickness of 0.8 mm, as measured with a position indicator (NP37, Iskra Avtomatica, Ljubljana, Slovenia) coupled with a stylus surface profilometer. Two rectangular LIG sensor devices were fabricated on commercial polyimide tape (Kapton^®^, DuPont, DE, USA) using a 1 cm × 2 cm active graphene area and transferred onto PDMS/PEG. This geometry was selected to ensure sufficient mechanical deformation under finger bending while maintaining a sufficient contact area for wire attachment. LIG was transferred onto the PDMS/PEG surface using a stamping method, which involves manual mechanical pressure. Conductive copper foils with wires were attached to the sides of LIG. The wires were connected to a Keysight 34461A, which was interfaced with a desktop computer. Measurements were performed in the constant current mode with the current set to 1 mA, and the voltage was measured over several minutes. The experimental setup is depicted in Figure 2. Devices were retested after 3 weeks of storage in air, during which no significant changes in resistance or signal shape were observed.

## 3. Results and Discussion

### 3.1. Polymer Synthesis

A series of four PDMS composites with different PEG content ranging from 10 wt.% to 40 wt.% was synthesized by hydrosilylation, i.e., the addition reaction of silicon hydrides (Si-H) across double bonds (e.g., vinyl groups). The two components included component A, a PDMS base containing vinyl groups, and component B, a curing agent with Si-H bonds. A chemically cross-linked polymer was formed. Photographs of synthesized pure PDMS and PDMS/PEG composites are shown in Figure 3. Pure PDMS was transparent, while PDMS/PEG composites were opaque white after the polymerization process.

After synthesis, PDMS/PEG composites were irradiated with a CO_2_ laser, resulting in laser-induced graphene. The scheme of fabrication of LIG is shown in Figure 4.

### 3.2. Electrical Characterization of LIG on PDMS/PEG

Laser parameters were optimized to investigate their influence on the properties of LIG. The optimal laser parameters were those that yielded (i) continuous and crack-free LIG patterns, (ii) the lowest electrical resistance, and (iii) optimal Raman spectra. The resolution was set to 1200 DPI, as it was empirically confirmed to be the optimal resolution for LIG fabrication on PDMS/PEG. Laser power for irradiation of PDMS/PEG was in the range of 15–18% of the 60 W maximum (9–10.8 W), and laser speed was varied between 35 and 55 mm s^−1^. A power above 10.8 W caused substrate degradation and graphene with destroyed edges, whereas graphene induction did not occur below a power of 9 W. A speed greater than 55 mm s^−1^ resulted in LIG that was visually assessed to be of low quality and had high electrical resistance, whereas with a speed lower than 35 mm s^−1^, there was no induction of LIG. Tested laser parameters on PDMS/PEG with fixed laser resolution 1200 DPI and different PEG content are shown in Appendix A.

Photographs of LIG on PDMS with different PEG content are shown in Figure 5a–d, corresponding to PEG content between 10 and 40 wt.%. Photographs marked from 1 to 3 correspond to laser parameters with a fixed scanning speed of 45 mm s^−1^, resolution of 1200 DPI, and laser power of 10.8 W, 9.6 W, and 9 W, respectively. Photographs marked from 4 to 6 correspond to laser parameters with a fixed laser power of 9 W, fixed resolution of 1200 DPI, and scanning speed of 35, 45, and 55 mm s^−1^, respectively.

The electrical resistance of LIG/PDMS/PEG with fixed laser speed 45 mm s^−1^ and resolution 1200 DPI and LIG/PDMS/PEG with fixed laser power 9 W and resolution 1200 DPI are presented in Appendix A. LIG that was induced on PDMS with 40 wt.% of PEG appeared darkest, which is a visual indication that this is the optimal concentration of PEG for laser induction of graphene. Measurements of electrical resistance presented in Appendix A showed that LIG induced on PDMS with 40 wt.% of PEG had the lowest value of 500 Ω sq^−1^ in comparison with other materials. From visual appearance, it was hypothesized that graphenization possibly occurred on materials with 10 and 20 wt.% PEG content, but the electrical resistance was unstable, and the material did not have the black coloration expected for LIG.

A higher PEG content (50 wt.%) resulted in samples that were soft and mechanically unstable for practical applications. The black coloration of the irradiated surface served only as a preliminary indicator of carbonization; however, visual inspection alone is not sufficient to confirm graphene formation.

### 3.3. Raman Spectroscopy

Figure 6 depicts the most representative Raman spectra of LIG on PDMS substrates with varying PEG content (10–40 wt.%). All spectra correspond to LIG produced with 9 W of laser power, a resolution of 1200 DPI, and a scanning speed of 45 mm s^−1^. Close-up sections of 2D bands are depicted in Appendix A. An additional set of Raman spectra was recorded for LIG produced with 9 W of laser power, a resolution of 1200 DPI, and a scanning speed of 55 mm s^−1^ (Appendix A). Positions of D, G, and 2D bands, and the ratio of the intensity of the D to the G peaks and 2D to the G peaks were analyzed, together with crystallite size (*L_a_*) and full-width half maximum (FWHM) are shown in Appendix A, respectively.

Raman spectra of LIG on PDMS with different PEG content (10–40 wt.%) (Figure 6) show three prominent peaks that are typical features of graphene, as summarized in Appendix A. A strong G band is observed at ~1585 cm^−1^ originating from a first-order zone-boundary phonon, a D band at ~1348 cm^−1^ occurs due to defects, vacancies, and bent sp^2^ carbon bonds, and a 2D band at ~2700 cm^−1^ originates from second-order zone-boundary phonons [8,50]. The ratio of the intensity of the D to the G peaks and 2D to the G peaks was analyzed, together with *L_a_* and FWHM (Appendix A), so the crystallite size and the degree of graphenization could be studied (Figure 7).

We have used spectrum deconvolution to obtain *I_D_/I_G_*, *I*_2*D*_/*I_G_*, FWHM, and *L_a_* (Figure 7a–c, respectively), fitting the Raman spectra to five functions (three Voigt and two Gaussian) for D and G bands (Appendix A). Three Voigt functions were ascribed to G, D, and D’ bands, and two Gauss functions were ascribed to D* and D” peaks [63,66]. For 2D and D + G bands, we fitted the Raman spectra to Voigt and Gauss functions, respectively (Appendix A). The value was calculated as the average value from the three Raman spectra recorded. The ratio of the intensity of the 2D to the G peaks increases from 0.1 to 0.9 with the increase in PEG content, and in all cases, is smaller than 1, which indicates multilayered LIG [50,67]. Our findings are consistent with the literature data, which include Raman analysis of LIG formed on other precursors [6,50,66]. Numerous Raman spectroscopy studies of LIG indicate similar spectral positions of the G peak, I_D_/I_G_ ratios, and FWHM of all peaks as in our case, which corroborate the expectation of defective, few-layer LIG [9].

FWHM is a measure of structural disorder. The high FWHM values for the G and D peaks at PEG contents of 10 wt.% and 20 wt.% therefore indicate amorphous or disordered material (Figure 7b) [68]. As the PEG content increased to 40 wt.%, the FWHM of these peaks decreased, confirming a reduction in structural disorder and the formation of more crystalline graphene domains. The narrowing of the FWHM is consistent with the observed trends and confirms the improved quality of the graphene at higher PEG content.

The crystallite size was calculated from Equation (1) and shown in Figure 7c. It shows that the size increases with increasing PEG content from 7 to 17.6 nm. In particular, LIG on PDMS/40% PEG had twice the crystallite size compared to the other materials, suggesting the formation of better defined graphene domains [3]. Previous studies concluded that a high ratio of the intensity of the D peak to the G peak indicates that there are smaller crystal domains in graphene [50,69]. The lowest ratio between the intensity of the D peaks and the G peaks was found for LIG on PDMS with 40 wt.% of PEG with a value of 1.1, indicating the lowest concentration of defects and the highest crystallite size (*L_a_* = 17.6 nm) [49,70]. These results indicate that PEG plays an important modifier role in laser induction and improves the quality of graphene.

PEG probably promotes graphenization through its decomposition behavior under laser irradiation. Its carbon-rich, oxygen-containing structure generates volatile by-products that facilitate localized temperature peaks and graphitic restructuring. In addition, reduced cross-linking and increased chain mobility in PEG-rich regions enable better energy absorption and pyrolysis efficiency.

### 3.4. FTIR Spectroscopy

FTIR spectra of pure PDMS and LIG fabricated on PDMS/PEG composites with varying PEG content (10–40 wt.%) are depicted in Figure 8. LIG was produced with laser power set to 9 W, a scanning speed set to 45 mm s^−1^, and resolution set to 1200 DPI. The assignment of bands to these spectra is presented in Appendix A. The FTIR spectrum of a select PDMS/PEG composite is shown in Appendix A, and the assignment of bands in that spectrum is shown in Appendix A.

The broad absorption peak at around 3400 cm^−1^ corresponds to stretching vibrations of the O-H group. It is noticeable that the lowest intensity of the O-H stretching band is seen in the spectrum of LIG on PDMS/40% PEG. Such a band was reported to be an indication of LIG formation on other polymer substrates, such as PDMS/Triton composites, for example [50].

The bands at ~2900 cm^−1^ and 2960 cm^−1^, corresponding to asymmetrical stretching vibrations of CH_2_, are present in all spectra, implying the presence of PDMS/PEG. The absorption peak at ~1631 cm^−1^ is associated with the stretching vibration of the C=C group and is a strong indication of graphene formation. The spectrum of LIG on PDMS with 40 wt.% of PEG contains an absorption peak at 1445 cm^−1^, which correlates to the deformation asymmetric vibration of the C-H group. In the FTIR spectrum of pure PDMS, we can see two bands at 1054 and 1010 cm^−1^ associated with Si-O-Si vibrations. In the FTIR spectra of LIG materials, these two bands merge into a single peak at ~1070 cm^−1^, indicating the formation of agglomerated SiO_2_ nanoparticles due to the decomposition of PDMS under the high temperatures of CO_2_ laser irradiation [13]. The absorption peak at around 800 cm^−1^ indicates that Si-C is formed due to the thermal degradation of the PDMS matrix, which is also confirmed with XRD [36].

### 3.5. SEM

SEM was used to characterize the structure of laser-induced graphene visually. We sputtered a thin layer of copper onto the samples prior to imaging to enhance conductivity. Figure 9 depicts SEM micrographs of LIG on the surface of PDMS/PEG composites with PEG content ranging from 10 to 40 wt.%. A cross-section of LIG on PDMS/40% PEG is depicted in Appendix A. For these micrographs, LIG was produced with 9 W of laser power, a resolution of 1200 DPI, and a scanning speed of 45 mm s^−1^. All samples contain the typical porous structure of LIG along with few-layer flakes [3,5,71]. Using the application Image-Pro Plus 6.0, the diameter of the pores on the surfaces of different LIG/PDMS/PEG materials was measured (Appendix A). The average pore diameter for every material was presented in the Appendix A, showing that the smallest pores were found on LIG/PDMS/40% PEG with the average value of 0.88 µm. We employed EDX to analyze the elemental composition of LIG. The analysis revealed the presence of carbon, silicon, and oxygen in our samples, as depicted in Figure 10 and Appendix A. High oxygen concentration in all materials could be related to the oxidation of LIG in air during laser processing [72].

### 3.6. XRD

X-ray diffraction analysis was performed on LIG powder scraped from the substrate and on LIG films still on the substrate. The diffractograms for the powder and the films with different PEG content are shown in Figure 11a,b. The diffractograms of the PDMS/PEG composites without the graphene are shown in Appendix A.

In the diffractograms of the LIG powder scraped from the substrate, peaks at ~22.06° and ~41° can be seen, which is close to the peak positions at ~26° and ~43° expected for the graphene corresponding to the (002) and (100) planes (Appendix A), confirming the successful induction of graphene [5,73]. A diffraction maximum at ~35° is associated with β-SiC crystallites, which intensifies as the PEG content is reduced [37,74]. Using the Bragg equation n λ = 2 d sinθ, the in-plane spacing was calculated to be 3.93 Å (first order diffraction n = 1, X-ray wavelength λ = 1.54 Å and an angle of incidence of 11.29°), which is consistent with literature data [36,50,75].

In the LIG films on PDMS/PEG, the diffraction peaks characteristic of graphene were also positioned at slightly lower angles than reported in the literature, most likely due to interference with the substrate materials [50]. The diffraction peak at 26°, which corresponds to the graphene peak, overlaps with an amorphous halo at 22°, which is characteristic of PEG. The amorphous halo at 12° was attributed to PDMS [49]. The intensity of the diffraction maximum at 41°, which is characteristic of graphene, decreases with increasing weight concentration of PEG in PDMS. The diffraction maximum at ~35° is associated with the (111) in-plane structure of the β-SiC crystallite, indicating laser modifications of PDMS in SiC [36,74].

### 3.7. Swelling Behavior

The degree of swelling in toluene (*q_e_*), the crosslinking density (*v*) and the average molar mass by the number of polymer chains between the crosslinking sites (*M_c_*) for different PEG contents (10–40 wt.%) were derived to quantify the structural properties of the polymer composites. The dependence of these values on the PEG content is shown in Figure 12. Polymers that have network structures, such as PDMS, can swell to a certain degree depending on the solvent and structure of the polymer [64].

The results show that an increasing PEG content significantly influences the network properties of the composites. The degree of swelling in toluene increased with increasing PEG content, reflecting the lower cross-linking density of the materials. This behavior is attributed to the fact that PEG interferes with the hydrosilylation reaction during curing and hinders the formation of a tightly cross-linked PDMS network. Accordingly, the crosslink density decreased, while Mc increased with higher PEG content. The lower crosslink density in materials with higher PEG content is attributed to the inhibition of the PDMS crosslinking reaction by PEG [76].

While the sample with 40 wt.% PEG clearly showed higher swelling and lower cross-linking density, the difference between the samples with 20 wt.% and 30 wt.% PEG was less pronounced. This non-linear trend indicates that other factors are responsible for the swelling behavior in addition to the cross-linking density. These include (i) the polarity mismatch between PEG and the non-polar solvent toluene, which may reduce the swelling efficiency at intermediate PEG concentrations [48]; (ii) possible microphase separation between PEG and PDMS, leading to structural inhomogeneities; (iii) the presence of loosely bound PEG chains that are not fully integrated into the network and can absorb solvents without contributing to crosslink density; and (iv) changes in chain mobility and free volume caused by PEG as a plasticizer. Together, these factors contribute to the fact that the relationship between crosslink density and swelling is not simply inverse, and should be taken into account when interpreting the structural properties of PDMS/PEG composites.

### 3.8. Mechanical Characteristics and Environmental Stability

The mechanical properties of the materials were investigated by means of tensile tests in which the elongation at break, the modulus of elasticity and the tensile strength were determined. The materials tested were pure PDMS and PDMS/PEG composites.

It can be seen that with an increase in the PEG content in the PDMS matrix, the elongation at break increases from 95% to 237%. This increased elasticity can be attributed to the plasticizing effect of PEG, which reduces the intermolecular interactions within the PDMS matrix and thus enables greater flexibility. Conversely, the modulus of elasticity decreased significantly with increasing PEG content, reaching its lowest value of 0.25 MPa for the PDMS/40 wt.% PEG composite. This decrease in stiffness is a result of the weakened cross-linked network created by the PEG interference during PDMS curing. The measured modulus of elasticity (Young’s modulus) is consistent with literature data [36,50]. The elongation at break, modulus of elasticity and tensile strength for all materials are plotted in Figure 13. PDMS/40% PEG has the highest elasticity of all the materials investigated.

Tensile strength exhibited a non-linear trend. The addition of 10 wt.% PEG caused small changes, while the addition of 20 wt.% PEG resulted in a twofold increase in tensile strength. At lower PEG contents (e.g., 10–20 wt.%), intermolecular hydrogen bonding or dipole–dipole interactions between PEG chains and siloxane segments of PDMS may temporarily enhance the cohesive strength of the network without significantly disrupting cross-linking. This could lead to localized reinforcement, resulting in a transient increase in tensile strength before the softening and plasticization effects dominate at higher PEG contents. However, further increases in PEG content led to a decline in tensile strength, likely due to excessive softening of the matrix. Namely, as PEG content increases beyond 20 wt.% PEG, interference with the hydrosilylation reaction becomes dominant, leading to a more loosely cross-linked network and a drop in tensile strength, consistent with a classical softening behavior.

The mechanical performance, particularly the enhanced elongation and flexibility of PDMS/PEG composites compared to values reported for other siloxane materials and composites, underscores the potential for using PDMS/PEG composites in stretchable and wearable electronic devices [13,36,40,46]. For example, the elongation at break measured here is significantly larger than that observed in PDMS/PI composites [47].

To test the durability, environmental stability and cross-sensitivity of our sensors to temperature and humidity, we exposed the devices to controlled temperature and humidity conditions and measured the electrical resistance. Subsequently, we performed mechanical testing on the samples that were exposed to these conditions. Detailed results are shown in Appendix A. Electrical resistance is stable across a range of humidity levels for all devices (with maximum fluctuation of 8%). This suggests excellent long-term stability under variable humidity, crucial for wearable devices exposed to sweat or ambient moisture. The devices also show no clear sensitivity to temperature, with the exception of the device with 40 wt.% PEG, which exhibits an increase in resistance with rising temperature. The rise in resistance with temperature could be due to mechanical stretching of PEG, which has a positive thermal expansion coefficient. In standard sensor engineering practice, such temperature dependence is often compensated using external electronics. Following exposure to temperatures up to 50 degrees and humidities up to 90%, the mechanical properties of the substrate materials slightly degrade, although they remain highly favorable for applications in wearable devices.

### 3.9. Water Contact Angle and Water Absorption

To evaluate the hydrophilicity of samples, the water contact angles for pure PDMS, PDMS/PEG, and LIG on PDMS/PEG were measured with a goniometer. Photographs of water droplets on the PEG-containing materials are depicted in Figure 14. A photograph of a water droplet on pure PDMS is presented in Appendix A.

The results for pure PDMS and PDMS/PEG (Figure 14i) show a trend of decreasing water contact angle from 103.6° to 37.8° with increasing PEG content, which shows increasing hydrophilicity with added PEG. A particularly pronounced change is observed with the addition of 40 wt.% PEG, which was expected because PEG itself is hydrophilic.

After laser induction of graphene on PDMS/PEG, the surface becomes hydrophobic for substrates with 10 to 30 wt.% PEG and superhydrophobic for the substrate with 40 wt.% PEG. In this case, the water droplets form spheres with very low adhesion to the surface and easily roll off the surface (Appendix A) [77]. A water contact angle of more than 150° indicates a superhydrophobic surface [77,78,79]. Due to the induction of LIG, the contact angle increases from 141° to 154° the higher the PEG content. Although PEG is hydrophilic, our results show that graphene dominates the wetting behavior of these surfaces. With increasing PEG content, the quality of the graphene increases, leading to higher contact angles. The material with 40 wt.% PEG content is superhydrophobic.

The results presented in Figure 14j show that water absorption increased from 13.6% to 44.5% with increasing PEG content. These results are in agreement with the observed increase in hydrophilicity, further confirming the impact that PEG has on water uptake in the composites.

### 3.10. XPS

XPS was performed to investigate the surface chemical composition of LIG on the PDMS/PEG composite with 40 wt.% of PEG. Figure 15 depicts XPS survey spectra that reveal the presence of C, O, and Si. Elemental composition determined with XPS was 37.2 at.% of C, 34.9 at.% of O, and 27.9 at.% of Si, which is in agreement with results from EDX. The presence of silicon and oxygen is attributed to the formation of silicon dioxide (SiO_2_) nanoparticles on the LIG surface due to the thermal degradation of PDMS. Additionally, oxygen can also be present due to oxidation that occurs during laser induction. Figure 16 depicts high-resolution XPS spectra showing the deconvolution of C 1s, O 1s, and Si 2p peaks. The C 1s spectrum (Figure 16a) was deconvoluted into five peaks: C=O (286.6 eV), C-O (285.7 eV), C-C (284.9 eV), C=C (284.2 eV), and Si-C (282.5 eV). The sp^2^ carbon hybridization content was 44%, indicating the presence of LIG, and the C=O content was 26.6%, indicating that oxidation of graphene has occurred, which is in correlation with the high oxygen content observed with EDX. Similar ratios of oxygen content and sp^2^ carbon bonds were found in previous work on LIG [20,34,50,72]. Figure 16b represents the O 1s spectrum deconvoluted into three peaks: C-O (533.6 eV), C=O (532.9 eV), and Si-O (531.9 eV). The carbonyl functional group C=O content was 66.7%, which can also be an indicator that oxidation of graphene occurred in the oxygen atmosphere [34,50,72]. A high-resolution Si 2p spectrum is shown in Figure 16c and deconvoluted into two peaks: Si-O (104 eV) and Si-C (102.7 eV). The percentage of Si-C was 91.9%, indicating that laser modification of PDMS to Si-C occurred, which is in agreement with XRD results.

### 3.11. TEM

TEM was performed on LIG powder scraped from a PDMS/PEG substrate with 40 wt.% PEG and dispersed in ethanol. The TEM micrographs are depicted in Figure 17. A low-magnification TEM micrograph (Figure 17a) reveals typical few-layer graphene. A higher magnification image, shown in Figure 17b, reveals parallel lines that indicate diffraction from few-layer graphene. The image in the inset of Figure 17b is obtained by taking the Fast-Fourier Transform (FFT) of the main image. The inverse Fourier transform of the sidelobe region reveals an interlayer spacing of ≈3.51 Å, corresponding to graphene interlayer spacing in the (002) orientation [37]. These results are in agreement with previously reported information about LIG structure and characteristics, where the inter-planar spacing was 3.4 Å [5,34].

### 3.12. TG Results in Nitrogen

The thermal stability and degradation behavior of LIG/PDMS/PEG materials were investigated using thermogravimetric analysis (TG) and derivative thermogravimetry (dTG). The measurements were carried out under a nitrogen atmosphere from room temperature to 700 °C. The onset of degradation (clay), the residual masses at 650 °C and the peak temperatures of degradation (*T_max_*) are summarized in Table 1 and the corresponding TGA and dTG curves are shown in Figure 18a,b, respectively.

The onset temperature of degradation (*T_on_*) decreases significantly with increasing PEG content (Figure 18a). The *T_on_* for LIG/PDMS/10% PEG was 418.2 °C, while for 20%, 30%, and 40% PEG, it dropped to 413.2 °C, 343.0 °C, and 342.6 °C, respectively (Table 1). This decline is attributed to the increased proportion of PEG, which is less thermally stable than PDMS and undergoes earlier degradation due to the cleavage of ether bonds in the PEG chains.

The residual mass at 650 °C (Figure 18a, Table 1) reflects the quantity of thermally stable components remaining after pyrolysis. Interestingly, the highest residue was observed for the LIG/PDMS/30% PEG sample (12.3%), followed closely by the samples with 10% and 40% PEG (11.6% and 10.6%, respectively). The lowest residue (7.8%) was recorded for the LIG/PDMS/20% PEG. These variations suggest that the interaction between PEG and PDMS significantly affects the carbonization yield, which is relevant for LIG formation. The relatively high residue for 30 wt.% PEG suggests that this composition provides an optimal balance between thermal decomposition of PEG and stabilization of char-forming species, supporting efficient carbonization. The residue for 40 wt.% PEG, although slightly lower than 30 wt.% PEG, still indicates sufficient carbonaceous yield for successful laser-induced graphenization, in agreement with Raman and XPS findings.

Derivative thermogravimetry (dTG) curves (Figure 18b) provide insight into the individual degradation events by highlighting the temperatures at which maximum mass loss occurs (*T_max_*; Table 1). All materials showed at least two major degradation peaks. The first peak (*T*_*max*1_), assigned to PEG decomposition, shifts significantly depending on composition. For example, *T*_*max*1_ is 180.8 °C in the LIG/PDMS/10% PEG sample but increases to 279 °C for 20% PEG, increases again for 30% PEG (304 °C), and for 40 wt.% PEG (299 °C). This behavior of the 30% and 40% PEG samples may reflect differences in PEG dispersion and interactions with the PDMS matrix, resulting in delayed decomposition of the PEG segments.

The main decomposition peaks associated with PDMS degradation appear in the range of 443–491 °C. For instance, the second degradation peak (*T*_*max*2_), typically assigned to depolymerization and cyclic oligomer formation from PDMS, appears at 448 °C (10% PEG), 443 (20 wt.% PEG), 447 (30 wt.% PEG) and 448 °C (40 wt.% PEG), consistent with the plasticizing and crosslinking-inhibiting effects of PEG. The third degradation peak (*T*_*max*3_), associated with residual backbone degradation and char formation, is observed near 490 °C for the 10% PEG sample.

Taken together, the thermal analysis confirms that increasing PEG content compromises the thermal stability of the LIG/PDMS/PEG materials but may facilitate the formation of carbonaceous residue, which is critical for efficient laser-induced graphenization. The reduced onset degradation temperature and altered thermal decomposition pathways indicate that PEG acts as both a plasticizer and a carbon source and supports its role as a carbon source facilitating graphenization during laser irradiation.

### 3.13. Limb Motion Sensor

Due to its excellent piezoresistive properties, LIG was used as a limb motion sensor. LIG was fabricated on a PDMS/PEG composite with 40 wt.% PEG. Graphene was induced on the surface of polyimide and then transferred to PDMS/PEG. Induction on polyimide with subsequent transfer was performed because this method significantly improved the adhesion of the electrical contacts compared to LIG induced directly on PDMS/PEG. To detect finger flexion, the sensor was attached to the finger with adhesive tape, as shown in the photo in Figure 19. The relative change in resistance during limb movement was calculated as Δ*R/R*_0_, where Δ*R* and *R*_0_ are the change in resistance due to finger movement and the initial resistance, respectively. In such a measurement, the value of Δ*R*/*R*_0_ is zero when the resistance returns to the initial value, and larger values indicate that the resistance changes due to limb movement. The relative resistance as a function of time is shown in Figure 19 for an example of 2 min. The measurements were taken over a total of 10 min with the thinner substrate, which has a lower thickness (800 µm). The measurements taken with the thicker substrate are shown in Appendix A. The resistance increased when the finger was bent and returned to its initial value when the finger was straightened. The finger was bent and straightened at a constant frequency, which is reflected in the signal obtained. The sensor did not lose its conductivity even when the finger was fully flexed (angle 95°) (Appendix A). The device is sensitive enough to distinguish between small and large finger flexion movements.

Furthermore, to evaluate the sensitivity of the sensor, the gauge factor was calculated using Equation (6) after a maximum 10% strain on a Universal Testing Machine (Appendix A), yielding a value of 347. The response time of our sensor was 200 ms, as determined by measuring finger bending over a 20 s interval, with the mean value calculated from multiple sudden finger bending motions (Appendix A), at a measured bending angle of ~50° (Appendix A). Table 2 shows a comparison between our devices and other LIG-based sensors on PDMS substrates. The response time of our sensors is short, and the Gauge factor is high (Table 2 and Appendix A).

In our study, LIG/PDMS/PEG with 40 wt.% PEG exhibited the lowest I_D_/I_G_ (1.1) and highest *L_a_* (17.6 nm), indicating the fewest structural defects. Such material is expected to have enhanced electrical conductivity, which is essential for achieving a stable piezoresistive signal in the sensor. Furthermore, SEM analysis (Appendix A) revealed a decrease in average pore size with added PEG (~0.88 µm for 40 wt.% PEG), which is expected to provide a mechanically stable, yet conductive network that responds effectively to deformation. In contrast, samples with higher defect density and larger pore size (e.g., 10 wt.% PEG, with an average pore diameter of 3.77 µm) exhibited smaller elongation at break and higher Young’s modulus, with a smaller gauge factor, making them less useful for applications in strain and bending sensing. These results demonstrate a direct link between micro-nano structure and device-level performance: higher graphitic order and finer porosity enable better electro-mechanical performance.

Compared with existing LIG-based strain sensors, our PDMS/PEG-based sensor offers several distinct advantages [20,80,81]. The incorporation of 40 wt.% PEG into the PDMS matrix enabled direct or transferred laser graphenization, improved substrate elasticity (elongation at break of 237%), and facilitated the formation of high-quality few-layer graphene with minimal defects. The resulting micro-nano structured LIG exhibits a porous morphology and high crystallinity, leading to a gauge factor of 347 and a response time of 200 ms under small strains (up to 10%), outperforming many reported systems [20,80,81]. The surface of the LIG/PDMS/PEG composite is superhydrophobic (154° contact angle), which enhances device robustness in humid or sweat-exposed environments. This combination of mechanical flexibility, chemical biocompatibility, high piezoresistive sensitivity, and environmental resilience positions our sensor as a promising candidate for next-generation wearable electronics.

**Table 2 sensors-25-05238-t002:** Gauge factor, cycles, and response time of LIG/PDMS-based sensors in this work and other research papers.

	Gauge Factor; Strain Range	Cycles	Response Time (ms)	References
LIG on PDMS/PEG	347; *ε* = 0–10%	450	200	this work
LIG on pure PDMS	41; *ε* = 30–46%	12,000	250	[82]
LIG on pure PDMS	111; *ε* = 0–1.6%	1500	1040	[25]
LIG on pure PDMS	43; *ε* = 45–48%	5000	300	[83]
LIG on pure PDMS	15.8; *ε* = 10–20%	2000	160	[20]
LIG on pure PDMS	1242 (n.l.); *ε* = <25%	12,000	250	[84]

The long-term stability of our sensors was demonstrated by performing finger bending tests over 1000 cycles (a period of 20 min). The data for sensors on PDMS/PEG materials with 20 wt.%, 30 wt.%, and 40 wt.% of PEG are shown in Appendix A. There is no significant drift or hysteresis over the measured period, except for a small drift in the case of 20 wt.% PEG, which indicates very good repeatability and sensor stability. We performed additional gauge factor measurements on all three materials. The gauge factors listed in Appendix A indicate that the sensitivity of the sensors is consistently high, especially for the material with 40 wt.% PEG.

Although the technical challenge of affixing contacts to our devices based on direct LIG/PDMS/40% PEG remains, there were a small number of cases when contacts adhered to the samples sufficiently to make measurements. One such case is shown in Appendix A. Although the signal from the sensor was intermittent, the sample was conducting and reacted to finger bending. Appendix A also depicts the detachment of contacts, a technical challenge to be solved in the future before practical application of our devices can be realized. While we observed occasional contact detachment, these results confirm that direct LIG on PDMS/PEG can serve as a standalone sensing layer, though further optimization of contact interfaces is needed for long-term device deployment.

Our results demonstrate the potential of LIG-based sensors on PDMS/PEG for advanced applications in motion monitoring. The ability to detect small body movements makes these sensors suitable for use in healthcare, robotics, and remote gesture control. Furthermore, the biocompatibility, flexibility, and piezoresistive sensitivity of the materials highlight their potential for integration into wearable electronics.

The use of LIG transferred from PI to PDMS/PEG was motivated by the current limitations in contact adhesion on directly induced LIG. This approach allowed us to evaluate sensor performance in realistic conditions, such as finger bending and long-term cycling. The sensor responses observed with transferred LIG confirm that the underlying PDMS/PEG composite supports stable and reproducible piezoresistive behavior and serves as a viable substrate for both directly and indirectly patterned graphene structures. The findings validate the functionality of the substrate and support its future use with directly induced LIG once contact interface optimization is achieved.

## 4. Conclusions

In this work, we demonstrate the successful fabrication of LIG on a novel biocompatible PDMS/PEG substrate and its application as a sensitive and flexible strain sensor for finger flexion detection. Comprehensive physicochemical investigations were carried out using electrical resistivity measurements, Raman spectroscopy, FTIR spectroscopy, SEM-EDX, XRD, TEM and XPS, all of which confirmed the successful fabrication of the LIG on the polymer substrate.

The incorporation of PEG into the PDMS matrix improved the flexibility and hydrophilicity, which was reflected in an increased elongation at break, a reduced modulus of elasticity and a reduced water contact angle. The elongation at break of PDMS with a PEG content of 40 wt.% reached 237%, exceeding the values reported for other LIG/PDMS-based materials [13,49]. The addition of PEG appears to have inhibited the crosslinking of PDMS, making the composites more extensible and hydrophilic. Measurements of the water contact angle showed the superhydrophobicity of LIG. After laser induction of graphene, the surface became superhydrophobic, with a contact angle of 154° for LIG on PDMS/40 wt.% PEG, highlighting the dominant role of graphene in adjusting the wetting behavior of the surface. The comprehensive physicochemical characterization performed in this work revealed critical insights into the structure–function relationship in LIG/PDMS/PEG sensors. Raman spectroscopy demonstrated that higher PEG content leads to larger sp^2^ crystallite domains, fewer defects, and improved graphenization, which directly enhanced electrical conductivity and piezoresistive performance. FTIR and XRD analyses confirmed the formation of graphene, SiC, and SiO_2_ phases, indicating successful high-temperature carbonization and PDMS degradation. SEM and TEM imaging revealed a porous, few-layered graphene architecture. EDX and XPS analyses confirmed the chemical composition and surface oxidation state of the LIG, with XPS revealing a high sp^2^ carbon content and the presence of oxygen functionalities, both of which influence interfacial stability and electrical transport. Together, these techniques support a unified understanding of how microstructure, chemical composition, and morphology govern the sensitivity, stability, and mechanical properties of the final sensor device.

XPS, Raman, and TEM analyses confirmed the formation of few-layer graphene with a high proportion of sp^2^ carbon (44%), an expected quantity of defects for this type of graphene, and a crystallite size of up to 17.6 nm for PDMS/40 wt.% PEG. The Raman band positions and widths are in agreement with literature for LIG, as are the XPS-measured sp^2^ bond and oxygenation fractions. At the same time, XPS and EDX confirmed the presence of carbon, oxygen, and silicon, with oxygen related to surface oxidation. The formation of SiO_2_ nanoparticles and β-SiC crystallites indicated the thermal modification of PDMS during laser induction.

Adding PEG to the matrix decreases mechanical stability under extreme ambient conditions, such as temperatures of 50 degrees and humidity of 90%. The elongation at break decreases for high-PEG content samples that have been treated in extreme atmospheres. This feature highlights the tradeoff between improved electrical performance and reduced mechanical stability when adding PEG, although even the samples treated in extreme environments retain mechanical properties that are comfortably sufficient for use in wearable devices.

The fabricated LIG served as a conductive layer on the PDMS/PEG substrate for wearable limb motion sensors. The developed LIG-based limb motion sensor exhibited excellent piezoresistive performance and could effectively detect and discriminate finger movements. The biocompatibility, flexibility and robust performance of the LIG/PDMS/PEG composite suggest promising applications in healthcare, robotics and wearable technology.

The sensors showed reproducible resistance changes during bending and acceptable consistency between samples produced in different batches. The stability of the performance was dependent on the thickness of the substrate and the contact adhesion to the LIG layer.

Future research should focus on improving the adhesion of the electrical contacts to the LIG layer, optimizing and developing new biocompatible polymers as substrates for laser-induced growth, and integrating conductive polymers into the composites to improve their electrical and mechanical properties to further enhance their potential as motion sensors for precise and reliable applications.

## Figures and Tables

**Figure 1 sensors-25-05238-f001:**
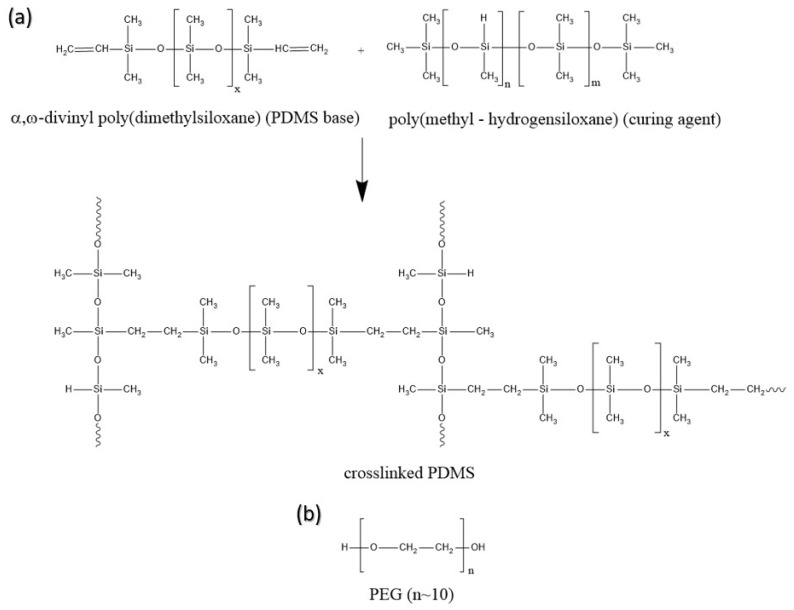
(**a**) Synthesis of cross-linked PDMS; (**b**) Structure of PEG.

**Figure 2 sensors-25-05238-f002:**
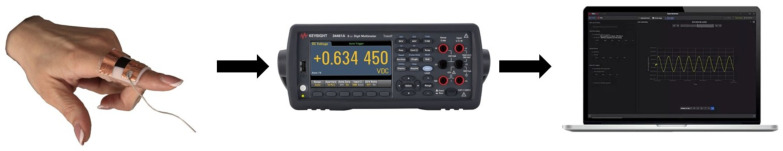
Schematic of the limb motion measurement process.

**Figure 3 sensors-25-05238-f003:**
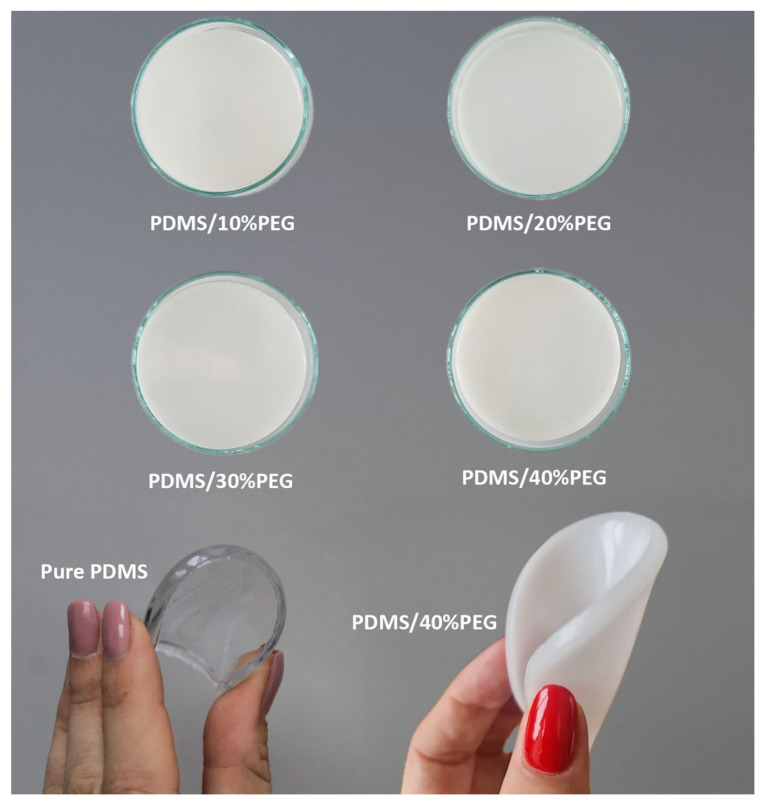
Photographs of PDMS/PEG composites with different PEG content (10–40 wt.%) and pure PDMS. All samples containing PEG are opaque white, whereas pure PDMS is transparent.

**Figure 4 sensors-25-05238-f004:**
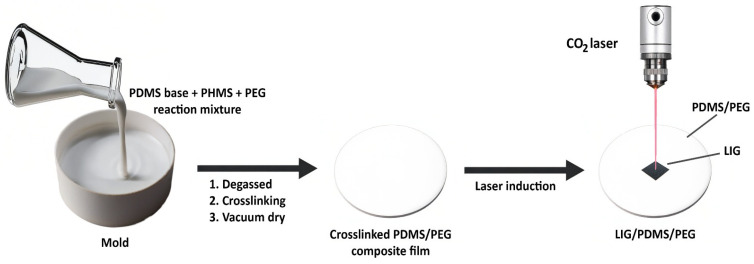
The scheme of polymer synthesis and LIG fabrication.

**Figure 5 sensors-25-05238-f005:**
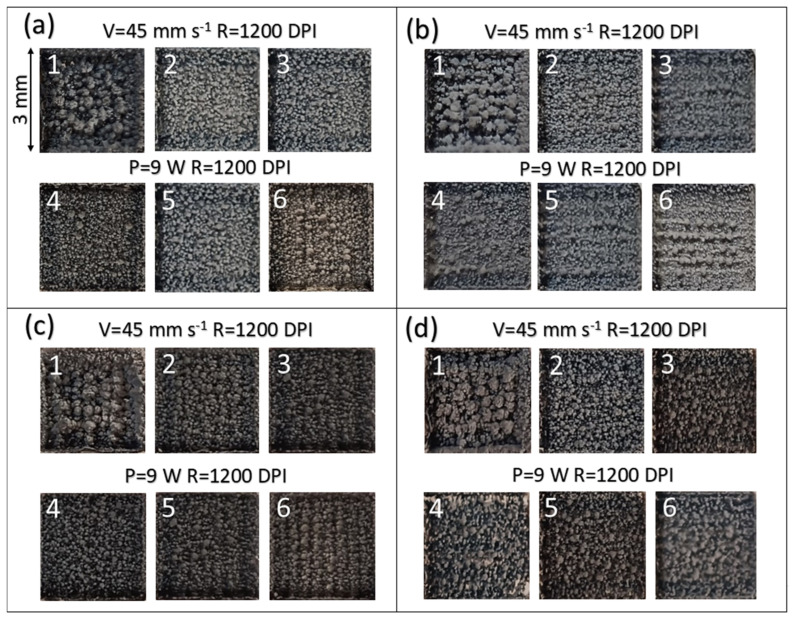
Photographs of laser-induced graphene on: (**a**) PDMS/10% PEG; (**b**) PDMS/20% PEG; (**c**) PDMS/30% PEG; (**d**) PDMS/40% PEG, made with (1–3) laser power 10.8 W, 9.6 W and 9 W, (4–6) scanning speed 35, 45 and 55 mm s^−1^. The length of the sides of squares depicted in the images is 3 mm.

**Figure 6 sensors-25-05238-f006:**
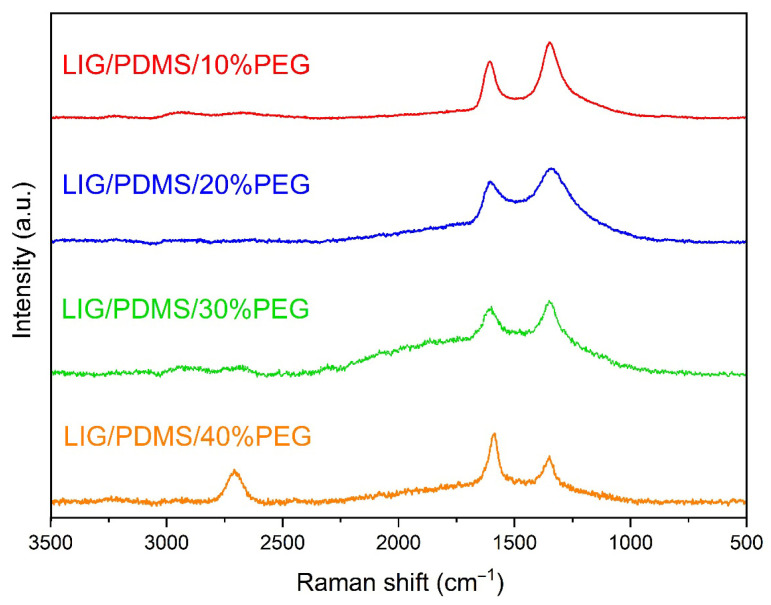
Raman spectra of LIG on PDMS with different PEG content (10–40 wt.%).

**Figure 7 sensors-25-05238-f007:**
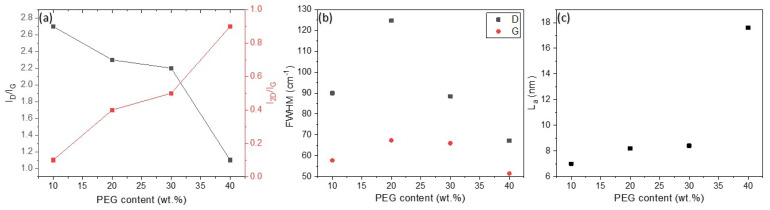
(**a**) Variations of *I_D_/I_G_* and *I*_2*D*_/*I_G_* versus PEG content; (**b**) variation in FWHM for the D and G band with PEG content; (**c**) variation of *L_a_* with PEG content.

**Figure 8 sensors-25-05238-f008:**
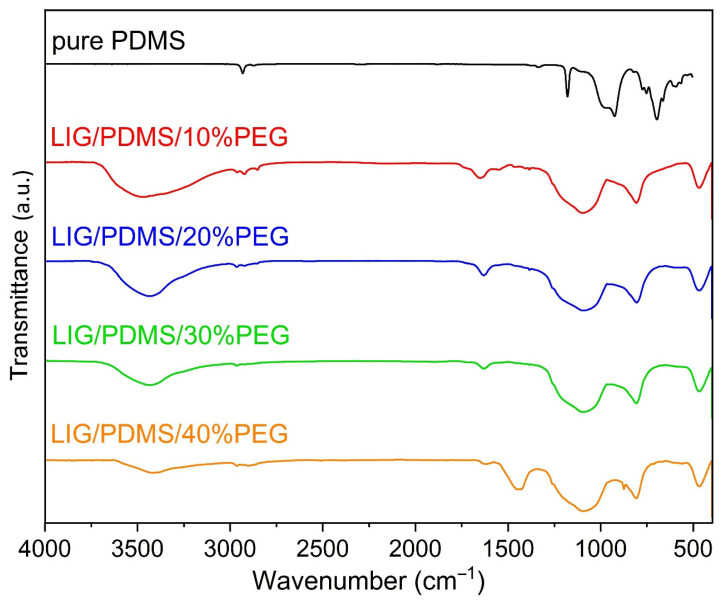
FTIR spectra of pure PDMS and LIG on PDMS/PEG with varying PEG content (10–40 wt.%).

**Figure 9 sensors-25-05238-f009:**
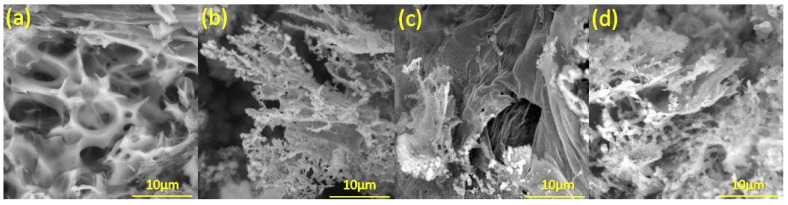
SEM micrographs of LIG on different substrates at magnification 5000×: (**a**) PDMS/10% PEG; (**b**) PDMS/20% PEG; (**c**) PDMS/30% PEG; (**d**) PDMS/40% PEG.

**Figure 10 sensors-25-05238-f010:**
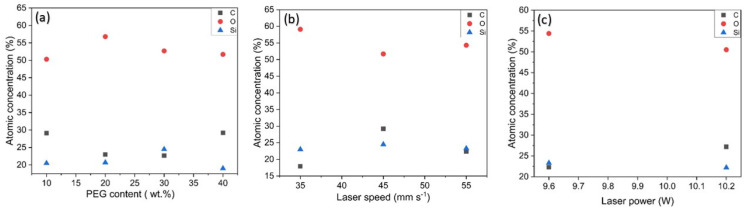
Atomic concentrations of different elements versus (**a**) PEG content, (**b**) laser speed, and (**c**) laser power.

**Figure 11 sensors-25-05238-f011:**
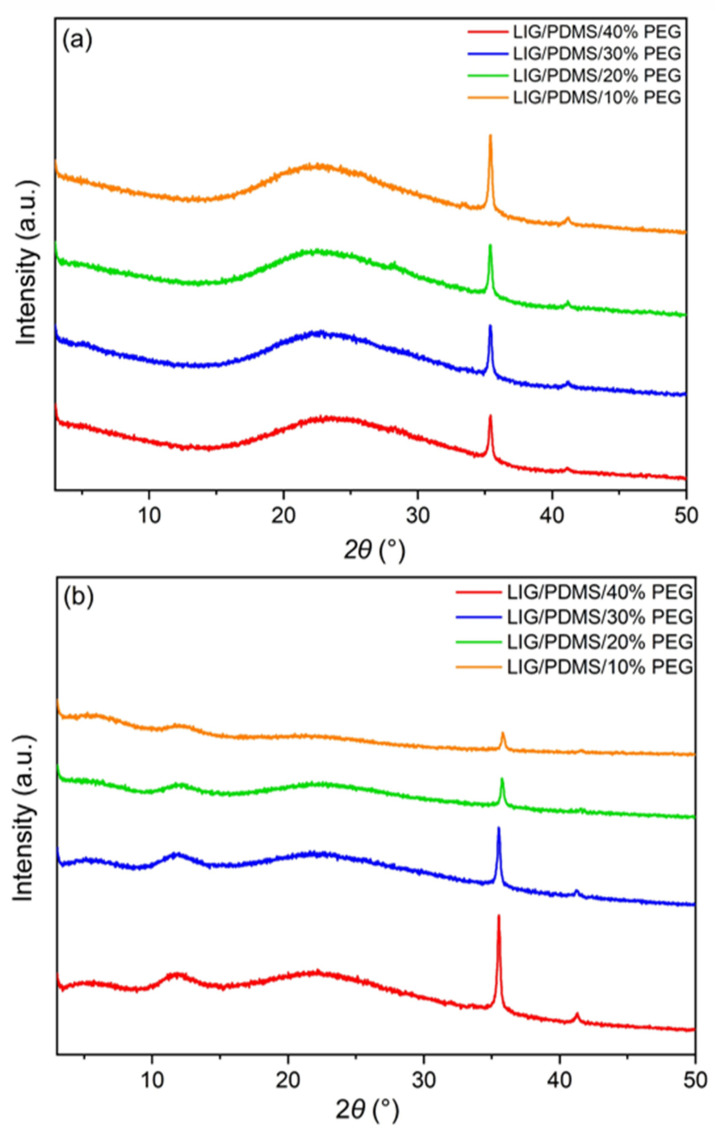
Diffractograms of (**a**) LIG powder and (**b**) LIG films with varying PEG content (10–40 wt.%).

**Figure 12 sensors-25-05238-f012:**
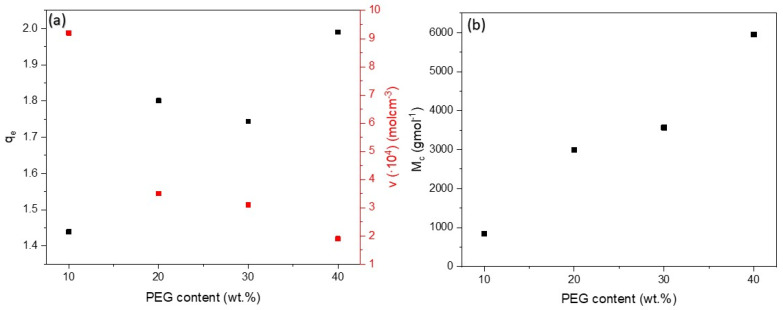
(**a**) Dependences of the degree of swelling (black squares) and cross-linking density (red squares) versus PEG content and (**b**) the average molar mass of the polymer chains between the cross-linking sites versus PEG content.

**Figure 13 sensors-25-05238-f013:**
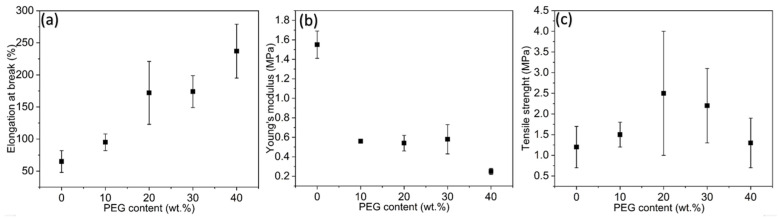
Variations in (a) elongation at break, (**b**) Young’s modulus, and (**c**) tensile strength versus PEG content in PDMS.

**Figure 14 sensors-25-05238-f014:**
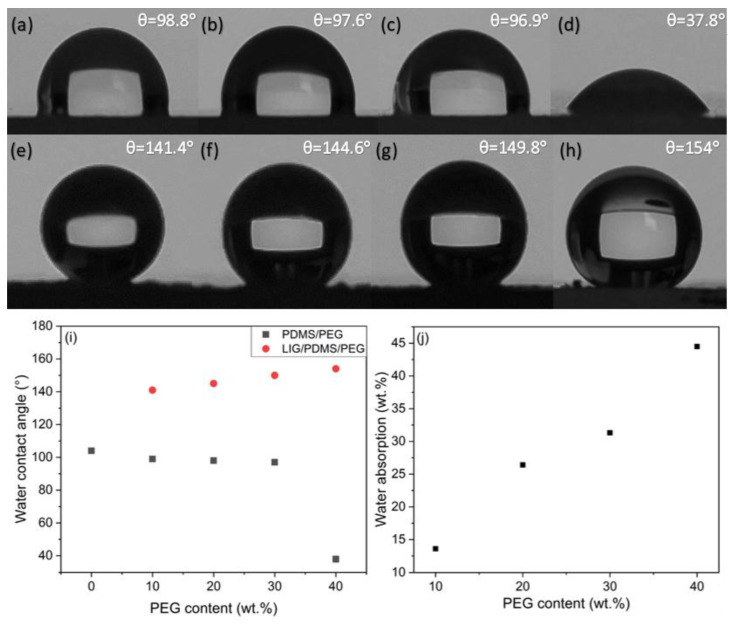
Photographs of droplets on PDMS with PEG content from 10 to 40 wt.% (**a**–**d**), and on LIG on PDMS with 10–40 wt.% PEG content (**e**–**h**); (**i**) relationship between water contact angle and PEG content for PDMS/PEG and LIG/PDMS/PEG materials and (**j**) relationship between water absorption and PEG content in PDMS/PEG composites.

**Figure 15 sensors-25-05238-f015:**
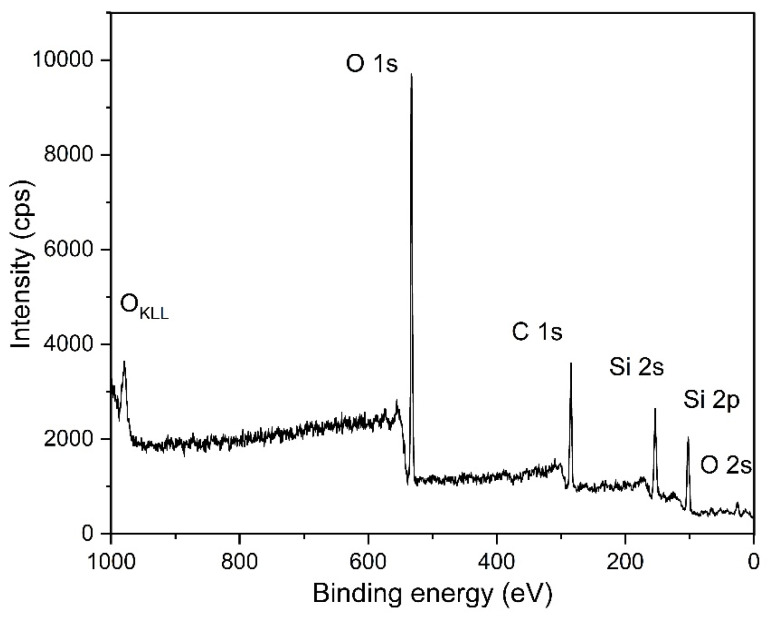
XPS survey spectra of LIG/PDMS/40% PEG.

**Figure 16 sensors-25-05238-f016:**
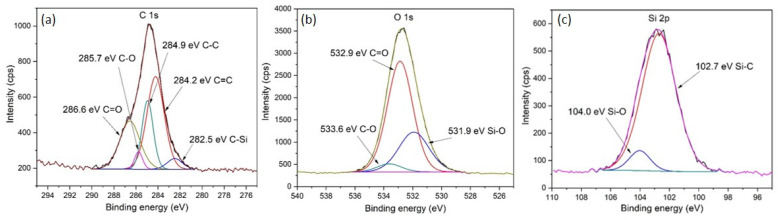
High-resolution XPS spectra of LIG/PDMS/40% PEG with deconvolution: (**a**) C 1s; (**b**) O 1s; (**c**) Si 2p. The differently colored curves represent different chemical bonds, as specified on the figure.

**Figure 17 sensors-25-05238-f017:**
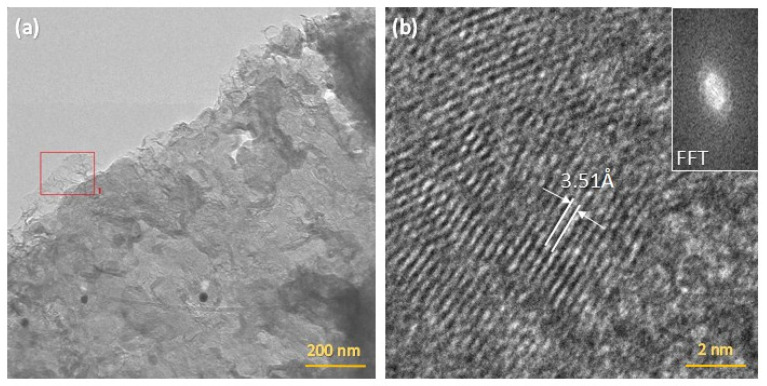
(**a**) TEM and (**b**) HRTEM micrographs with FFT (inset) of LIG/PDMS/40% PEG.

**Figure 18 sensors-25-05238-f018:**
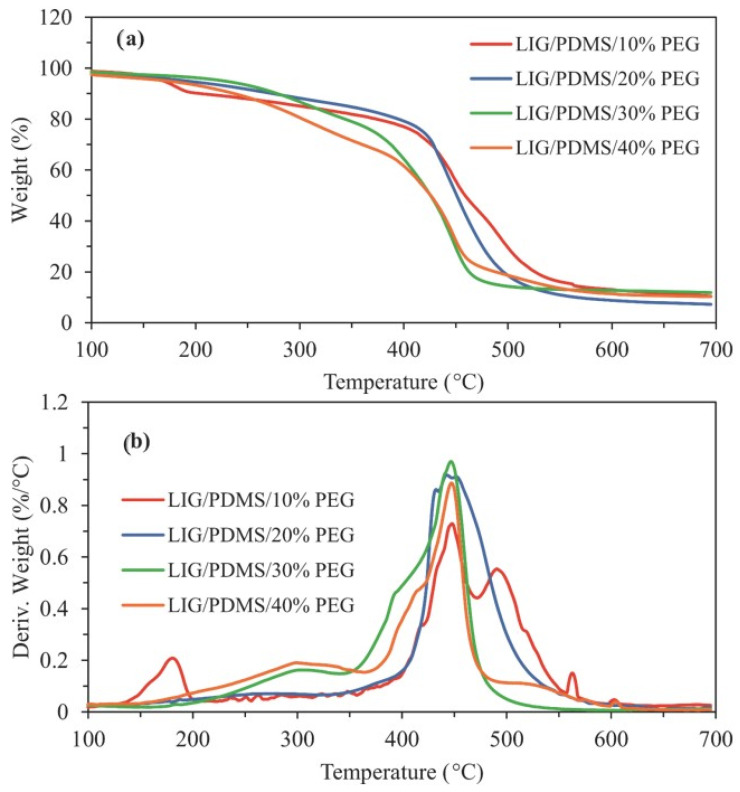
(**a**) TGA and (**b**) dTG curves of LIG/PDMS/PEG materials.

**Figure 19 sensors-25-05238-f019:**
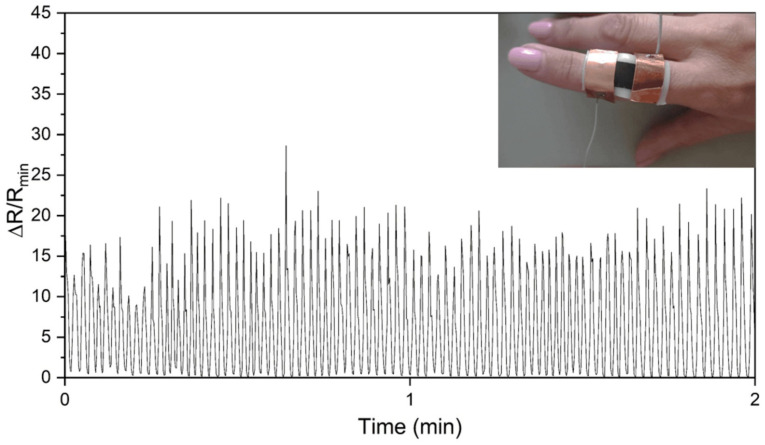
Relative resistance as a function of time in 450 cycles and a photograph of the tactile sensor (inset).

**Table 1 sensors-25-05238-t001:** TG/dTG results of LIG/PDMS/PEG materials.

Sample Code	*T_on_* (°C)	Residue at 650 °C (%)	*T_max_* (°C)
LIG/PDMS/10% PEG	418	11.6	181/448/491
LIG/PDMS/20% PEG	413	7.8	279/443
LIG/PDMS/30% PEG	343	12.3	304/447
LIG/PDMS/40% PEG	343	10.6	299/448

## Data Availability

Data is available upon request from the authors.

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
