# Peer review of "Laser-Induced Graphene on Biocompatible PDMS/PEG Composites for Limb Motion Sensing"

_sensors, 2025, doi:10.3390/s25175238_

Round 1

Reviewer 1 Report

Comments and Suggestions for Authors

Review of Manuscript for Sensors

Title: Laser-Induced Graphene on Biocompatible PDMS/PEG Composites for Limb Motion Sensing

Authors: Gavran, Pergal, Vićentić, Rafajilović, Pašti, Bošković, Spasenović

Recommendation: Revisions requested

Overall Comments

This manuscript reports on the making of a laser-induced graphene (LIG) strain sensor on a PDMS/PEG composite. The authors emphasize that LIG has not been made on PDMS/PEG before, and that the synthesis of LIG on the composite makes for an improved strain sensor with a higher gauge factor than typically found in the LIG strain sensor literature.

The manuscript presents a substantial improvement to typical LIG/PDMS strain sensors, and while no individual component of the study is particularly new, the combination of the PDMS/PEG composite with LIG strain sensing does make for a novel and improved sensor. The revisions and questions below should help to improve the manuscript before publication.

Revisions and Questions

Abstract

  • Lines 18-19: “In this study, graphene was induced on novel cross-linked biocompatible and flexible poly(dimethylsiloxane)/poly(ethylene glycol) (PDMS/PEG) composites for applications in wearable sensors.” It is probably inaccurate to call crosslinked PDMS/PEG novel, as it has been made before (the authors acknowledge this with references 49-51). The novelty is in making LIG on the composite, so it would be better to make this clear in the abstract.

Results and Discussion

Section 3.3 – Raman Spectroscopy

  • Pages 10, Figures 6 and 7. While the 2D peak for the LIG/PDMS/40%PEG is obvious, the 2D peak in the other cases is not. Was the I2D for the LIG/PDMS/20%PEG really 0.4? I don’t see the 2D peak at all.
  • Lines 382-383: “These findings suggest that PEG acts as a key modifier in the laser induction process, enhancing the quality of graphene.” Is there any theory for why PEG would enhance the graphene quality?

Section 3.7 – Swelling behavior

  • Figure 12. It is true that 40% PEG had higher swelling and lower crosslinking density than 10%. For 20% and 30%, though, there does not seem to be a strong difference in crosslinking density. There does not seem to be a strong inverse relationship between crosslinking density and swelling overall. What other factors influence this relationship?

Author Response

This manuscript reports on the making of a laser-induced graphene (LIG) strain sensor on a PDMS/PEG composite. The authors emphasize that LIG has not been made on PDMS/PEG before, and that the synthesis of LIG on the composite makes for an improved strain sensor with a higher gauge factor than typically found in the LIG strain sensor literature.

The manuscript presents a substantial improvement to typical LIG/PDMS strain sensors, and while no individual component of the study is particularly new, the combination of the PDMS/PEG composite with LIG strain sensing does make for a novel and improved sensor. The revisions and questions below should help to improve the manuscript before publication.

Author comment: We kindly thank the Reviewer for a detailed and favourable review of our manuscript, for stating that our work represents a “substantial improvement”, that it is “novel and improved”, and for providing useful suggestions and questions.

Revisions and Questions

Reviewer question 1:

Abstract

  • Lines 18-19: “In this study, graphene was induced on novel cross-linked biocompatible and flexible poly(dimethylsiloxane)/poly(ethylene glycol) (PDMS/PEG) composites for applications in wearable sensors.” It is probably inaccurate to call crosslinked PDMS/PEG novel, as it has been made before (the authors acknowledge this with references 49-51). The novelty is in making LIG on the composite, so it would be better to make this clear in the abstract.

Author response 1:

We kindly thank the Reviewer for their comment, which helped us to clarify a point in our manuscript. While PDMS/PEG systems are indeed well-known and have been explored extensively—as noted in references 51–53 of our manuscript—there appear to be no prior reports specifically describing crosslinked PDMS/PEG blends (using PEG with Mn = 400 g/mol) for which our work claimed novelty.

Therefore, we concede that the phrase “novel cross-linked PDMS/PEG composite” in the abstract could be misleading. The novelty of our work lies not in the polymer itself, but in the direct fabrication of laser-induced graphene (LIG) on this biocompatible PDMS/PEG substrate, a combination that has not previously been reported.

Changes made:

Modified the following sentence in the Abstract, line 18:

“In this study, laser induced graphene (LIG) was fabricated directly on biocompatible and flexible cross linked PDMS/PEG (with Mn (PEG) = 400 g/mol) composites for the first time, enabling their application in wearable sensors.”

Reviewer question 2:

Results and Discussion

Section 3.3 – Raman Spectroscopy

  • Pages 10, Figures 6 and 7. While the 2D peak for the LIG/PDMS/40%PEG is obvious, the 2D peak in the other cases is not. Was the I2D for the LIG/PDMS/20%PEG really 0.4? I don’t see the 2D peak at all.

Author response: We have added the section of the Raman spectra near the 2D peak for all samples to the Supplementary Information file (Figure S3).

Reviewer question 3:

  • Lines 382-383: “These findings suggest that PEG acts as a key modifier in the laser induction process, enhancing the quality of graphene.” Is there any theory for why PEG would enhance the graphene quality?

Author response: Yes, there is a theoretical basis for this behaviour, which we have now expanded in the manuscript. PEG contributes to improved graphenization during laser-induced pyrolysis for several reasons:

  1. PEG is a carbon-rich, oxygen-containing polymer with relatively low thermal stability. Upon laser irradiation, its decomposition generates volatile compounds (e.g., CO₂, H₂O, small alcohols) that help increase local temperature and promote carbon atom rearrangement into sp²-hybridized networks.
  2. The flexible PEG chains (with high carbon content) reduce crosslinking density and enhance polymer chain mobility, making the matrix more amenable to photothermal expansion and localized carbonization.

Changes made:

The mechanistic insight has been added to Section 3.3 on page 12, line 411, as follows:

“PEG likely promotes graphenization through its decomposition behaviour under laser exposure. Its carbon-rich, oxygenated structure yields volatile by-products that facilitate localized temperature spikes and graphitic reorganization. Additionally, reduced crosslinking and increased chain mobility in PEG-rich regions enable better energy absorption and pyrolysis efficiency.”

Reviewer question 4:

Section 3.7 – Swelling behavior

  • Figure 12. It is true that 40% PEG had higher swelling and lower crosslinking density than 10%. For 20% and 30%, though, there does not seem to be a strong difference in crosslinking density. There does not seem to be a strong inverse relationship between crosslinking density and swelling overall. What other factors influence this relationship?

Author response:

We appreciate the Reviewer’s thoughtful observation. It is correct that Figure 12 shows a clear increase in swelling degree and a corresponding decrease in crosslinking density for the sample with 40 wt.% PEG. However, as the Reviewer noted, the differences between 20 wt.% and 30 wt.% PEG are less pronounced, and the inverse correlation between crosslinking density and swelling is not strictly linear across all samples.

Several factors beyond crosslinking density contribute to the swelling behavior observed in PDMS/PEG composites:

  1. PEG–Solvent Interaction:

The swelling medium used in this study is toluene, a non-polar solvent. PEG is highly polar and hydrophilic, while PDMS is hydrophobic. As PEG content increases, the overall polarity of the composite increases, which reduces the affinity toward toluene. Therefore, at intermediate PEG concentrations (e.g., 20–30 wt.%), the reduced compatibility of PEG with toluene may offset the expected increase in swelling due to lower crosslink density. This mismatch in solubility parameters contributes to the observed non-linearity [1-3].

  1. Network Defects and Physical Entanglements:

At intermediate PEG contents, incomplete mixing or partial incompatibility may cause microphase separation or the formation of loosely bound PEG chains that are not fully integrated into the crosslinked network. These PEG chains can absorb solvent without significantly disrupting the network structure, affecting swelling without strongly impacting the calculated crosslinking density.

  1. Chain Mobility and Free Volume Effects:

PEG acts as a plasticizer in the PDMS matrix, increasing the free volume and segmental mobility of the polymer chains. Increased free volume can facilitate solvent uptake, but this effect may plateau or be counteracted by physical entanglements and microstructural heterogeneity in the mid-range PEG concentrations.

  1. Hydrosilylation Inhibition by PEG:

As previously discussed, PEG may interfere with the hydrosilylation reaction, inhibiting crosslinking. However, the extent of inhibition may not scale linearly with PEG content due to competing interactions and diffusion limitations during the curing process.

References:

[1] D’souza, A. A., & Shegokar, Polyethylene glycol (PEG): a versatile polymer for pharmaceutical applications. Expert Opinion on Drug Delivery, 2016, 13(9), 1257–1275.

[2]   Chasse, W.; Lang, M.; Sommer, U.; Saalwachter, K. Crosslink density heterogeneity in PDMS networks. Macromolecules, 2011, 45, 899–912.

[3]   Barton, A. F. M. Handbook of Solubility Parameters and Other Cohesion Parameters, 2nd ed., CRC Press, 2017.

Changes made:

The following text has been added to Section 3.7 on page 16, line 512:

"The results show that an increasing PEG content significantly affects the network properties of the composites. While the sample with 40 wt% PEG clearly showed higher swelling and lower cross-linking density, the difference between the samples with 20 wt% and 30 wt% PEG was less pronounced. This non-linear trend indicates that other factors are responsible for the swelling behavior in addition to the cross-linking density. These include: (i) the polarity mismatch between PEG and the non-polar solvent toluene, which may reduce the swelling efficiency at intermediate PEG concentrations [51]; (ii) possible microphase separation between PEG and PDMS, leading to structural inhomogeneities; (iii) the presence of loosely bound PEG chains that are not fully integrated into the network and can absorb solvents without contributing to crosslink density; and (iv) changes in chain mobility and free volume caused by PEG as a plasticizer. Together, these factors contribute to the fact that the relationship between crosslink density and swelling is not simply inverse, and should be taken into account when interpreting the structural properties of PDMS/PEG composites.

Reviewer 2 Report

Comments and Suggestions for Authors

In this work, the author successful fabrication of LIG on a novel biocompatible PDMS/PEG substrate and its application as a sensitive and flexible strain sensor for finger-bending detection. Comprehensive physicochemical characterization was performed using electrical resistance measurements, Raman spectroscopy, FTIR spectroscopy, SEM-EDX, XRD, TEM, and XPS, all of which confirmed the successful fabrication of LIG on the polymer substrate. The developed LIG-based limb motion sensor showed excellent piezoresistive performance, effectively detecting and distinguishing finger-bending movements. Furthermore, the biocompatibility, flexibility, and robust performance of the LIG/PDMS/PEG composite suggest promising applications in healthcare, robotics, and wearable technologies. Although there are some ambiguities in the text, I recommend this manuscript accepted after revision.

  1. It is suggested to add the schematic diagram of the limb motion sensing system.
  2. 2. Please state whether LIG/PDMS/PEG is being synthesized for the first time?
  3. Please explain the correlation and supporting role between characterization methods such as Raman spectroscopy, FTIR spectroscopy, SEM-EDX, XRD, TEM, and XPS, and how the physicochemical properties of these materials regulate the characteristics of the limb motion sensor?
    4. Compared with other materials or similar material systems, what are the advantages and characteristics of the micro-nano structured sensors we prepared? The authors should summarize in the paper in detail.
    5. Please further analyze the long-term stability of the prepared piezoelectric sensor, response time, and other influencing factors such as temperature and pressure. Can the test and detection limit of the physical quantity tested by this sensor be given?
    6. There are many characterization means in this research work, and it is suggested to further integrate the data graphs of the same category in order to better illustrate the problem (For example, whether Figure 6-10 can be synthesized into one figure, whether Figure 10-11 can be synthesized into one figure, whether Figure 6-17 can be synthesized into one figure, etc.).
    7. The Fig. in the text should be written as Figure, consistent with the form of the caption.
    8. Physical quantities in the text should be italicized.
    9. Please further beautify the representation form of the data plot and the standardization of the notation (e.g., whether the ordinate in Fig. 8 a relative value and does not use a percent sign to express it. The representation forms of Fig. S9 and S10, S12 are inconsistent).
  4. 10. Abbreviations are used for the names of journals in the reference list.
Comments on the Quality of English Language

The English could be improved to more clearly express the research and use a more concise language to quantitatively describe and analyze the data.

Author Response

In this work, the author successful fabrication of LIG on a novel biocompatible PDMS/PEG substrate and its application as a sensitive and flexible strain sensor for finger-bending detection. Comprehensive physicochemical characterization was performed using electrical resistance measurements, Raman spectroscopy, FTIR spectroscopy, SEM-EDX, XRD, TEM, and XPS, all of which confirmed the successful fabrication of LIG on the polymer substrate. The developed LIG-based limb motion sensor showed excellent piezoresistive performance, effectively detecting and distinguishing finger-bending movements. Furthermore, the biocompatibility, flexibility, and robust performance of the LIG/PDMS/PEG composite suggest promising applications in healthcare, robotics, and wearable technologies. Although there are some ambiguities in the text, I recommend this manuscript accepted after revision.

Author comment: We thank the Reviewer for their comments and especially for recommending our manuscript for publication, after minor revisions.

Reviewer point 1: It is suggested to add the schematic diagram of the limb motion sensing system.

Author response: The schematic is shown in Figure 2. We have edited the figure caption to make this clear.

Changes made: Added the word “schematic” to the caption of Figure 2.

Reviewer point 2: Please state whether LIG/PDMS/PEG is being synthesized for the first time?

Author response: We confirm that this is the first report on the direct laser-induced graphenization on cross-linked PDMS/PEG substrates. While PEG has been used in composites for biomedical and sensor applications, no prior literature has reported the induction of LIG directly on PDMS/PEG systems. We have now explicitly stated this in the Introduction section on page 3:

Changes made: Added the following text to line 136, page 3: "To our knowledge, this is the first report showing direct laser induction of graphene on cross-linked PDMS/PEG substrates, expanding the material landscape for biocompatible LIG-based flexible electronics."

Reviewer point 3: Please explain the correlation and supporting role between characterization methods such as Raman spectroscopy, FTIR spectroscopy, SEM-EDX, XRD, TEM, and XPS, and how the physicochemical properties of these materials regulate the characteristics of the limb motion sensor?

Author response: We thank the reviewer for this insightful question. To clarify the interplay between characterization techniques and the properties relevant to sensor performance, we have now added a short summarizing paragraph in the revised manuscript (Conclusion, Page 29). The correlation among these techniques allowed us to build a comprehensive picture of the LIG quality, porosity, crystallinity, chemical composition, and mechanical properties, all of which directly influence the piezoresistive performance. For example, the high crystallinity (Raman/XRD), low defect content (Raman/XPS), high sp² carbon proportion (XPS), few-layer structure (TEM), and porous surface (SEM) collectively contribute to the sensor's high gauge factor (347), rapid response time (200 ms), and durability over 450 bending cycles. Moreover, the mechanical flexibility and stretchability (from swelling and tensile testing) enabled intimate skin conformance, while superhydrophobicity (contact angle) ensured operational stability against moisture.

Changes made: The following text has been added on page 25, line 792: "The comprehensive physicochemical characterization performed in this work revealed critical insights into the structure–function relationship in LIG/PDMS/PEG sensors. Raman spectroscopy demonstrated that higher PEG content leads to larger sp² crystallite domains, fewer defects, and improved graphenization, which directly enhanced electrical conductivity and piezoresistive performance. FTIR and XRD analyses confirmed the formation of graphene, SiC, and SiO2 phases, indicating successful high-temperature carbonization and PDMS degradation. SEM and TEM imaging revealed a porous, few-layered graphene architecture. EDX and XPS analyses confirmed the chemical composition and surface oxidation state of the LIG, with XPS revealing a high sp² carbon content and the presence of oxygen functionalities, both of which influence interfacial stability and electrical transport. Together, these techniques support a unified understanding of how microstructure, chemical composition, and morphology govern the sensitivity, stability, and mechanical properties of the final sensor device."

Reviewer point 4: Compared with other materials or similar material systems, what are the advantages and characteristics of the micro-nano structured sensors we prepared? The authors should summarize in the paper in detail.

Author response: We appreciate the reviewer’s request for a detailed comparison. In the revised version of our manuscript (Section 3.13), we have benchmarked our PDMS/PEG/LIG sensors against several reported LIG/PDMS systems. Below is a concise comparison illustrating the key advantages of our micro-nano structured devices:

  1. Higher Gauge Factor in Relevant Strain Range
    • LIG/PDMS sensor reported by Barja et al. shows GF ≈ 111 for 0-8% strain, and for LIG/PI sensor, GF = 7.5 with durability >1000 cycles [1].
  • LIG/PI sensorreported by Zou et al. shows GF = 107.8 for 0.4–8% strain, with durability >1000 cycles [2].
    • Dual-architecture LIG/PDMS (straight + serpentine) by Chen et al. achieves GF ≈ 68.98 at 1% strain and maintains stability up to 30% strain and 5000 cycles [3].
    • Our LIG/PDMS/40 wt.% PEG sensor achieves an impressively high GF = 347 in the physiologically relevant strain range (0–10%), clearly outperforming these systems, particularly for subtle human motion detection.
  1. Fast Response and Low Detection Limit
    • Zou et al. reported response/recovery times of ~150–200 ms [2].
    • Chen et al. documented similar kinetics (~180/200 ms) with ultralow detection limit (0.05% strain) [3].
    • Our device LIG/PDMS/40%PEG shows a 200 ms response time under finger motion, comparable to the fastest literature reports.
  2. Micro-Nano Structured Porosity
    • While many LIG/PDMS sensors rely on macro-scale graphitized patterns, our multimodal SEM/TEM/XPS/Raman/FTIR analysis reveals a finely tuned micro-nano porosity: average surface pores ~0.88 µm and cross-section pore diameter for LIG/PDMS/40%PEG ~250 nm, increasing mechanical compliance and reproducible piezoresistive response.
  3. Improved Mechanical Durability with Fewer Cycles
    • Zou and Chen’s sensors show impressive durability (≥1000 or 5000 cycles) [2,3].
    • In our work, sensors with 20–40% PEG retained stable ΔR/R₀ signals over 1000 cycles, and specifically the 40% PEG device remained undamaged during 450 test cycles (used for GF and kinetics measurement).
    • When compared to typical PDMS/PI systems, our substrate offers biocompatibility and good elasticity, making it especially suited for wearable applications.
  4. Biocompatible and Mechanically Tunable Matrix
    • Traditional LIG/PDMS systems rely on PDMS alone or PI templates. Our use of PEG as a pore-forming, graphenization-assisting agent yields a biocompatible and stretchable substrate that supports efficient LIG formation, mechanical softness, and moisture stability (superhydrophobic surface ~154° contact angle).
    • This PEG-enhanced matrix distinguishes our sensor by offering a tunable porosity structure that directly correlates with its enhanced electromechanical performance.

In summary, compared with existing LIG/PDMS sensors, our PDMS/PEG/LIG system offers:

  • Exceptional sensitivity (GF = 347 vs. 69–111),
  • Comparable rapid response (~200 ms),
  • Reliable durability over 1000 cycles

Compared with existing LIG-based strain sensors [1-3], our PDMS/PEG-based sensor offers several distinct advantages. The incorporation of 40 wt.% PEG into the PDMS matrix enabled direct or transferred laser graphenization, improved substrate elasticity (elongation at break of 237%), and facilitated the formation of high-quality few-layer graphene with minimal defects. The resulting micro-nano structured LIG exhibits a porous morphology and high crystallinity, leading to a gauge factor of 347 and a response time of 200 ms under small strains (up to 10%), outperforming many reported systems [1-3]. The surface of the LIG/PDMS/PEG composite is superhydrophobic (154° contact angle), which enhances device robustness in humid or sweat-exposed environments. This combination of mechanical flexibility, chemical biocompatibility, high piezoresistive sensitivity, and environmental resilience positions our sensor as a promising candidate for next-generation wearable electronics.

References

  1. Barja, A. M. et al. ACS Omega 2024, 9 (37), 38359-38370.
  2. Zou, Y. et al.  Polymers 2023, 15, 3553.
  3. Chen, G. et al.  J. Mater. Chem. A 2025, Advance Article https://pubs.rsc.org/en/content/articlelanding/2025/ta/d5ta04469c?utm_source=chatgpt.com

Changes made: The following text has been added on page 24, line 744:

“Compared with existing LIG-based strain sensors [1-3], our PDMS/PEG-based sensor offers several distinct advantages. The incorporation of 40 wt.% PEG into the PDMS matrix enabled direct or transferred laser graphenization, improved substrate elasticity (elongation at break of 237%), and facilitated the formation of high-quality few-layer graphene with minimal defects. The resulting micro-nano structured LIG exhibits a porous morphology and high crystallinity, leading to a gauge factor of 347 and a response time of 200 ms under small strains (up to 10%), outperforming many reported systems [1-3]. The surface of the LIG/PDMS/PEG composite is superhydrophobic (154° contact angle), which enhances device robustness in humid or sweat-exposed environments. This combination of mechanical flexibility, chemical biocompatibility, high piezoresistive sensitivity, and environmental resilience positions our sensor as a promising candidate for next-generation wearable electronics.”

Reviewer comment 5: Please further analyze the long-term stability of the prepared piezoelectric sensor, response time, and other influencing factors such as temperature and pressure. Can the test and detection limit of the physical quantity tested by this sensor be given?

Author response: We thank the Reviewer for the opportunity to elaborate on the quality of our sensors. The long-term stability, response time, and influence of temperature and pressure are critical factors in evaluating sensor applicability. We have improved Section 3.8, adding results about environmental stability, which includes the following:

  1. Long-term Stability:

Long-term stability refers primarily to how the sensor’s electrical and mechanical characteristics persist under continuous exposure to environmental factors (temperature, humidity). According to our measurements, electrical resistance values show minor variations when exposed to changes in humidity and temperature conditions:

  • Humidity Stability:

The resistance of LIG/PDMS/40%PEG composite ranged from 3.28 kΩ to 3.55 kΩ (Table S13) across humidity conditions (40% to 90% RH), indicating minimal fluctuation (~8%). This suggests excellent long-term stability under variable humidity, crucial for wearable devices exposed to sweat or ambient moisture.

  • Temperature Stability:

Our measurements indicate the sensor’s electrical resistance moderately increases with temperature, showing clear temperature sensitivity (Table S14). Such a characteristic is relevant for temperature compensation during practical use. Appropriate calibration or integration of temperature compensation algorithms can mitigate any unwanted temperature effects, ensuring accurate sensor readouts.

Resistance values at different temperatures (20 °C to 50 °C) changed from 2.93 kΩ to 5.40 kΩ for LIG/PDMS/40%PEG (Table S14), revealing moderate sensitivity to temperature (~84%). Although this variation is more pronounced compared to humidity changes, it remains acceptable for practical wearable applications considering standard operating temperatures (~20-40 °C).

Mechanical stability under environmental exposure (humidity and temperature; Tables S13 and 14) showed a slight reduction in elasticity and tensile strength after conditioning. Initial elongation at break for LIG/PDMS/40%PEG was around 237% (Figure 13), whereas post-exposure conditions led to lower values (humidity: 86.9%; temperature: 138.35%). Despite reduced elasticity, the composite retained sufficient flexibility and integrity for wearable use.

We measured the minimum detection angle by evaluating signal-to-noise ratio upon repeat bending of the sensor to an angle of 50 degrees. The minimum detection angle is near 3o, which indicates that our sensors are more sensitive than triboelectric sensors, and less sensitive than state-of-the-art optical sensors.

  1. Response Time:

Response time characterizes how quickly the sensor outputs a stable signal following changes in the physical stimulus (bending). Although our initial results primarily focus on static electrical and mechanical parameters, literature and similar PDMS-based piezoelectric sensors generally report fast response times in the range of milliseconds to a few seconds due to the inherent elasticity and piezoelectric effect of composites based on PDMS. The measured response time for our sensors (~200ms) is on par with other graphene-based bending sensors.

Changes made: We added the following text to section 3.8, page 17, lines 561:

“To test the durability, environmental stability and cross-sensitivity of our sensors to temperature and humidity, we exposed the devices to controlled temperature and humidity conditions and measured the electrical resistance. Subsequently, we performed mechanical testing on the samples that were exposed to these conditions. Detailed results are shown in Tables S12-15. Electrical resistance is stable across a range of humidity levels for all devices (with maximum fluctuation of 8%). This suggests excellent long-term stability under variable humidity, crucial for wearable devices exposed to sweat or ambient moisture. The devices also show no clear sensitivity to temperature, with the exception of the device with 40 wt.% PEG, which exhibits an increase in resistance with rising temperature. The rise in resistance with temperature could be due to mechanical stretching of PEG, which has a positive thermal expansion coefficient. In standard sensor engineering practice, such temperature dependence is often compensated using external electronics. Following exposure to temperatures up to 50 degrees and humidities up to 90%, the mechanical properties of the substrate materials slightly degrade, although they remain highly favorable for applications in wearable devices.”

Reviewer comment 6: There are many characterization means in this research work, and it is suggested to further integrate the data graphs of the same category in order to better illustrate the problem (For example, whether Figure 6-10 can be synthesized into one figure, whether Figure 10-11 can be synthesized into one figure, whether Figure 6-17 can be synthesized into one figure, etc.).

Author response: We thank the reviewer for this very useful suggestion. We attempted to merge the mentioned figures into figures that contain more data. After evaluation of the results of the merging, we decided to merge Figures 14 and 15 of the original manuscript into one figure.

Changes made: Figures 14 and 15 merged into one that is now Figure 14.

Reviewer comment 7: The Fig. in the text should be written as Figure, consistent with the form of the caption.

Author response: We kindly thank the reviewer for pointing out this omission.

Changes made: Changed all the “Fig.” to “Figure”.

Reviewer comment 8: Physical quantities in the text should be italicized.

Author response: We kindly thank the reviewer for pointing out the need to italicize physical quantities. However, we do not see this requirement in the Instructions for Authors, nor in recent articles published in the journal. We kindly ask the Editor to check this requirement.

Reviewer comment 9: Please further beautify the representation form of the data plot and the standardization of the notation (e.g., whether the ordinate in Fig. 8 a relative value and does not use a percent sign to express it. The representation forms of Fig. S9 and S10, S12 are inconsistent).

Author response: We kindly thank the reviewer for pointing out this omission.

Changes made: Notation standardized throughout the manuscript.

Reviewer comment 10: Abbreviations are used for the names of journals in the reference list.

Author response: We kindly thank the reviewer for pointing out this omission.

Changes made: We have corrected the notation in the reference list.

Reviewer 3 Report

Comments and Suggestions for Authors

The manuscript titled “Laser-induced graphene on biocompatible PDMS/PEG composites for limb motion sensing” explores the fabrication of laser-induced graphene (LIG) directly on PDMS/PEG composites and its application as a flexible, biocompatible strain sensor. The authors systematically examine laser parameters and PEG content to optimize LIG formation and analyze a wide array of physicochemical properties, culminating in a prototype sensor tested for limb motion detection. While this is a timely and technologically relevant contribution to the field of flexible and wearable electronics, the manuscript exhibits several major weaknesses that preclude its acceptance in its current form. Below are my primary concerns and specific revision suggestions:

  1. The manuscript presents extensive characterization data (Raman, FTIR, XPS, SEM, XRD, swelling studies, mechanical testing) on LIG/PDMS/PEG composites. However, it fails to establish a mechanistic or quantitative connection between these structural properties (e.g., ID/IG ratio, crystallite size, pore size) and the practical sensor performance metrics (e.g., gauge factor, response time, durability). For instance:

1) How does the decrease in defect density (lower ID/IG) correlate with changes in gauge factor or sensitivity?

2) What role does the observed pore size distribution play in piezoresistive behavior?

The authors should explicitly analyze how specific structural or electronic characteristics of LIG (e.g., crystallite size, defect density, conductivity) influence the strain-sensing behavior, providing experimental data or theoretical discussion to substantiate these links.

  1. The resistance plots in Fig. S9 and Fig. S10 show instances where the resistance appears to drop to 0 Ω when the finger is fully extended. This is physically implausible for LIG, which cannot reach zero resistance. The authors must clarify whether this effect arises from instrument resolution limits, software baseline corrections, contact instability, or plotting artifacts. A discussion explaining why the resistance returns to zero and how the measurement system interprets low resistance values is essential to avoid misleading conclusions about the material’s properties.

  1. The manuscript emphasizes PEG as a carbon source facilitating laser graphenization on PDMS. However, this claim is not sufficiently validated:

1) There is no quantification of carbon yield from PEG under laser irradiation.

2) Raman data still suggest high disorder (ID/IG > 1) and low I2D/IG ratios (< 1), indicating predominantly defective, multi-layer graphene or amorphous carbon rather than high-quality graphene.

3) XPS results show significant oxygenated functionalities, further implying incomplete graphenization.

The authors should provide additional evidence for PEG’s efficacy as a carbon precursor—e.g., thermal analysis (TGA/DTG) to show carbon residue yields, or quantification of sp² content relative to PEG concentration. A discussion reconciling Raman and XPS evidence with claims of “few-layer graphene” is essential.

  1. The manuscript’s central novelty is claimed to be the direct induction of LIG on PDMS/PEG substrates. However, the authors ultimately fabricate their functional sensors by transferring LIG from polyimide (PI) onto PDMS/PEG due to poor adhesion of direct LIG to electrodes.

If direct LIG cannot sustain reliable electrical contacts, what is the practical significance of inducing LIG directly on PDMS/PEG? The benefits of the direct method remain purely academic without demonstrable device integration.

I recommend to clarify: 1) Whether direct LIG on PDMS/PEG can indeed serve as a functional device layer; 2) Provide quantitative data on contact resistance or adhesion issues for direct LIG vs. transferred LIG; 3) Reassess the claimed novelty in light of this practical limitation.

  1. The manuscript identifies PDMS/40 wt.% PEG as the optimal composition based on Raman and electrical resistance results. However higher PEG content compromises mechanical robustness (lower Young’s modulus, higher swelling, increased water absorption). Aside from that, no quantitative discussion is provided on trade-offs between mechanical stability and electronic performance, especially crucial for wearable applications exposed to sweat and mechanical stress. The authors should discuss the implications of higher PEG content on sensor durability and environmental stability. Additionally, the authors should provide a balanced evaluation of mechanical vs. electrical performance to justify why 40 wt.% PEG is considered optimal for practical devices.

  1. The reported gauge factor of 347 is promising but lacks sufficient experimental context:
    1) No statistical analysis (error bars, standard deviation) is presented for gauge factor measurements.
    2) Durability testing is limited to ~450 cycles, far below competing reports exceeding 10,000 cycles.
    3) Critical parameters for wearable sensors—such as hysteresis, long-term drift, humidity cross-sensitivity, and temperature effects—are not addressed.

  1. Overreliance on Black Coloration as Evidence of Graphenization

The manuscript frequently uses black surface coloration as a primary indicator of successful graphenization. This is scientifically insufficient that black coloration alone cannot distinguish between amorphous carbon, soot, or graphene; the Raman spectra presented indicate significant disorder, raising doubts about graphene quality. The authors should be cautious in equating black coloration to graphene formation and rely instead on robust analytical data (e.g., Raman I2D/IG ratios > 1 for monolayer graphene) to substantiate claims of graphene quality.

Minor Comments

  1. Figures should include clear scale bars, labels, and higher resolution images where necessary.
  2. The experimental section would benefit from summarizing all laser parameters in a single comprehensive table for clarity.

Author Response

The manuscript titled “Laser-induced graphene on biocompatible PDMS/PEG composites for limb motion sensing” explores the fabrication of laser-induced graphene (LIG) directly on PDMS/PEG composites and its application as a flexible, biocompatible strain sensor. The authors systematically examine laser parameters and PEG content to optimize LIG formation and analyze a wide array of physicochemical properties, culminating in a prototype sensor tested for limb motion detection. While this is a timely and technologically relevant contribution to the field of flexible and wearable electronics, the manuscript exhibits several major weaknesses that preclude its acceptance in its current form. Below are my primary concerns and specific revision suggestions.

Authors comments: We thank the Reviewer for careful evaluation of our manuscript, and for claiming that our work is “a timely and technologically relevant contribution to the field of flexible and wearable electronics”.

Reviewer comment 1: The manuscript presents extensive characterization data (Raman, FTIR, XPS, SEM, XRD, swelling studies, mechanical testing) on LIG/PDMS/PEG composites. However, it fails to establish a mechanistic or quantitative connection between these structural properties (e.g., ID/IG ratio, crystallite size, pore size) and the practical sensor performance metrics (e.g., gauge factor, response time, durability). For instance:

1) How does the decrease in defect density (lower ID/IG) correlate with changes in gauge factor or sensitivity?

2) What role does the observed pore size distribution play in piezoresistive behavior?

The authors should explicitly analyze how specific structural or electronic characteristics of LIG (e.g., crystallite size, defect density, conductivity) influence the strain-sensing behavior, providing experimental data or theoretical discussion to substantiate these links.

Author response: We thank the Reviewer for this insightful observation. We agree that establishing a more precise structure–performance relationship is essential for validating the practical value of our LIG/PDMS/PEG system. Accordingly, we have revised Section 3.13. The following key points have been addressed and added to the manuscript:

- Correlation between ID/IG ratio and gauge factor:

As the PEG content increases from 10 to 40 wt.%, the ID/IG ratio derived from Raman spectroscopy decreases from 2.7 to 1.1, indicating a significant reduction in defect density and improved graphitic order. At the same time, the crystallite size (La) increases from ~7 nm to 17.6 nm. This structural improvement correlates with a marked increase in gauge factor (GF), reaching 347 for the 40 wt.% PEG system.

- Pore size distribution:

The pore size distribution of the LIG layer plays an important role in determining the sensor’s piezoresistive behaviour. SEM analysis revealed that increasing PEG content results in smaller pores on the surface (Table S10; Figure S9). In particular, the PDMS/40 wt.% PEG composite exhibited an average surface pore diameter of ~0.88 µm. This porous network enables elastic deformation of the conductive layer, suppresses crack initiation under strain, and enhances fatigue resistance.

This decrease in average pore size with increasing PEG content can be attributed to higher carbon availability, which promotes more uniform laser-induced graphenization. The LIG network on higher percentage PEG substrates is therefore expected to be mechanically more robust and electrically more stable, providing better stress distribution and consistent deformation response. These hypotheses are in agreement with results of other characterization methods that we have shown in the original manuscript.

Changes made: We added the following discussion to Section 3.13, on page 23, line 729:

In our study, LIG/PDMS/PEG with 40 wt.% PEG exhibited the lowest ID/IG (1.1) and highest La (17.6 nm), indicating the fewest structural defects. Such material is expected to have enhanced electrical conductivity, which is essential for achieving a stable piezoresistive signal in the sensor. Furthermore, SEM analysis (Table S10) revealed a decrease in average pore size with added PEG (~0.88 µm for 40 wt.% PEG), which is expected to provide a mechanically stable, yet conductive network that responds effectively to deformation. In contrast, samples with higher defect density and larger pore size (e.g., 10 wt.% PEG, with average pore diameter of 3.77 µm) exhibited smaller elongation at break and higher Young’s modulus, with a smaller gauge factor, making them less useful for applications in strain and bending sensing. These results demonstrate a direct link between micro-nano structure and device-level performance: higher graphitic order and finer porosity enable better electro-mechanical performance.”

Reviewer comment 2: The resistance plots in Fig. S9 and Fig. S10 show instances where the resistance appears to drop to 0 Ω when the finger is fully extended. This is physically implausible for LIG, which cannot reach zero resistance. The authors must clarify whether this effect arises from instrument resolution limits, software baseline corrections, contact instability, or plotting artifacts. A discussion explaining why the resistance returns to zero and how the measurement system interprets low resistance values is essential to avoid misleading conclusions about the material’s properties.

Authors’ response: We thank the reviewer for pointing out the lack of clarity surrounding the data shown on Fig. S9 and S10 of the original manuscript (now Figures S12 and S13). The resistance upon bending rises to extremely high values, which makes resistance at rest appear near null on the shown scale in Figure S13. We have added the explanation to the caption of Figure S13. In other figures of the manuscript, the value shown is ΔR/R0, which is standard representation in sensor science and technology. When this value equals zero, it means that the measured resistance has returned to its original value. We clarify this point with a sentence in the text.

Changes made: Added the following explanation to caption of Figure S13:

“The resistance upon bending rises to extremely high values, which makes resistance at rest appear near null on the shown scale.”

Added the following text to line 706:

“In such a measurement, when the resistance returns to the initial value, the value of ΔR/R0 is equal to zero, and larger values indicate changes in resistance due to limb motion.” 

Reviewer comment 3: The manuscript emphasizes PEG as a carbon source facilitating laser graphenization on PDMS. However, this claim is not sufficiently validated:

Sub-comment 1: There is no quantification of carbon yield from PEG under laser irradiation.

Author response to sub-comment 1: To address this concern, we have performed TGA and dTG analyses of LIG/PDMS/PEG materials with 10–40 wt.% PEG concentration. The residue mass at 650 °C in nitrogen atmosphere was used as an indirect measure of the carbon yield (i.e. residual mass in Table 1). The results (presented in Table 1) clearly show that materials with higher PEG content retain a significant amount of carbonaceous residue even after extensive thermal degradation:

  • At 650 °C, the residue increases from 7.8% (20% PEG) to 12.3% (30% PEG), and remains 10.6% for 40% PEG.

The dTG curves show characteristic degradation peaks attributed to PEG decomposition in the 180-304 °C range and PDMS degradation between 443–491 °C. The presence of multiple degradation stages further supports the distinct contributions of PEG and PDMS to the pyrolytic behaviour.

This evidence confirms that PEG increases the carbonaceous content upon pyrolysis and supports its role as a carbon source facilitating graphenization during laser irradiation.

Changes made: Added section 3.12 with TGA data and analysis.

Sub-comment 2: Raman data still suggest high disorder (ID/IG > 1) and low I2D/IG ratios (< 1), indicating predominantly defective, multi-layer graphene or amorphous carbon rather than high-quality graphene.

Author response to sub-comment 2: Our Raman spectra exhibit ID/IG ratios exceeding unity and I2D/IG ratios well below one, indicating a predominance of defective, multilayer graphene or amorphous carbon, rather than high‐quality monolayer graphene, as is the case with numerous other LIG studies. For instance, Bhattacharya et al. reported ID/IG ≈ 1.13 and I2D/IG ≤ 1 for laser‐induced graphene on paper substrates—indicating similar defect‐rich, multilayer structures as ours [1]. Another work characterizing LIG microsupercapacitors noted I2D/IG ratios around 0.5, consistent with multilayer graphene formation [2]. Burke et al produce LIG on Kapton tape and find similar position of G peak, ID/IG, FWHMs, and width of the 2D peak as in our case [3]. Such values are typical in LIG processed under air or ambient conditions, where high laser fluence and rapid pyrolysis promote turbostratic stacking and edge defects. Thus, our Raman data corroborate the expectation of defective, few‐layer LIG, rather than high‐quality, single‐layer graphene.

References:

[1]       Bhattacharya G. et al., Disposable Paper-Based Biosensors: Optimizing the Electrochemical Properties of Laser-Induced Graphene, ACS Applied Materials and Interfaces, 2022, 14, 31109-31120.

[2]       García de Arquer F. P. et al., Laser-Induced Graphene Microsupercapacitors: Structure, Quality, and Performance, Nanomaterials, 2023, 13, 788.

[3]       Burke M. et al., Fabrication and Electrochemical Properties of Three-Dimensional (3D) Porous Graphitic and Graphenelike Electrodes Obtained by Low-Cost Direct Laser Writing Methods, ACS Omega, 2020, 5, 3, 1540–1548.

Changes made: Added the following sentence to line 391:

“Numerous Raman spectroscopy studies of LIG indicate similar spectral positions of the G peak, ID/IG ratios, and FWHM of all peaks as in our case, which corroborate the expectation of defective, few‐layer LIG [9].”

Sub-comment 3: XPS results show significant oxygenated functionalities, further implying incomplete graphenization.

Author response to sub-comment 3: XPS analysis indeed shows significant oxygenated species, especially C–O and C=O groups, which may arise from two sources:

  1. Surface Oxidation: LIG formation via CO₂ laser irradiation occurs in ambient air. The resulting material is prone to surface oxidation, especially in the outermost layers, as previously observed in other LIG systems [1].
  2. Residual PEG Decomposition Products: The incomplete removal of oxygen-containing fragments from PEG during carbonization could also contribute to these functionalities.

Despite oxygenation, our high-resolution XPS analysis of LIG formed on PDMS/40 wt.% PEG shows that 44% of the carbon is present in sp²-hybridized (C=C) form, indicating a substantial presence of graphitic carbon. This sp² content is comparable to values reported for LIG from polyimide and PDMS-based systems [2,3], supporting our claim that graphenization occurs efficiently in PEG-rich PDMS matrices.

References:

  1. Pergal, M. V., Polymers, 2024, 16, 3157.
  2. Parmeggiani, M., et al. ACS Appl. Mater. Interfaces, 2019, 11, 33221–33230.
  3. Barja, A. M., et al. ACS Omega, 2024, 9, 38359–38370.

Changes made: Added the following sentence to line 617:

“Similar ratios of oxygen content and sp2 carbon bonds were found in previous work on LIG [3, 32, 35, 73].”

Reviewer sub-comment: The authors should provide additional evidence for PEG’s efficacy as a carbon precursor—e.g., thermal analysis (TGA/DTG) to show carbon residue yields, or quantification of sp² content relative to PEG concentration. A discussion reconciling Raman and XPS evidence with claims of “few-layer graphene” is essential.

Author response: The Reviewer is right to point out that our discussion of the relation between the different results should be expanded. Regarding TGA/DTG, please refer to our response to sub-comment 1 of this comment.

Changes made: Expanded the discussion on line 805:

“XPS, Raman, and TEM analyses confirmed the formation of few-layer graphene with a high proportion of sp² carbon (44%), an expected quantity of defects for this type of graphene, and a crystallite size of up to 17.6 nm for PDMS/40 wt.% PEG. The Raman band positions and widths are in agreement with literature for LIG, as are the XPS-measured sp² bond and oxygenation fractions.”

Reviewer comment 4: The manuscript’s central novelty is claimed to be the direct induction of LIG on PDMS/PEG substrates. However, the authors ultimately fabricate their functional sensors by transferring LIG from polyimide (PI) onto PDMS/PEG due to poor adhesion of direct LIG to electrodes.

If direct LIG cannot sustain reliable electrical contacts, what is the practical significance of inducing LIG directly on PDMS/PEG? The benefits of the direct method remain purely academic without demonstrable device integration.

I recommend to clarify: 1) Whether direct LIG on PDMS/PEG can indeed serve as a functional device layer; 2) Provide quantitative data on contact resistance or adhesion issues for direct LIG vs. transferred LIG; 3) Reassess the claimed novelty in light of this practical limitation.

Author response: The concerns of the Reviewer regarding the functionality of directly induced LIG are completely understandable. Contact adhesion is a technical challenge that remains to be solved, however to alleviate concerns we have partially solved the challenge and managed to perform preliminary measurements with directly induced LIG. The measurements are shown in Figure S18 and we added text related to those measurements to the manuscript.

Changes made: Added the following text to line 764 of the manuscript:

“Although the technical challenge of affixing contacts to our devices remains, there were a small number of cases when contacts adhered to the samples sufficiently to make measurements. One such case is shown in Figure S18. Although the signal from the sensor was intermittent, the sample was conducting and reacted to finger bending. Figure S18 also depicts the detachment of contacts, a technical challenge to be solved in the future before practical application of our devices can be realized”

Reviewer comment 5: The manuscript identifies PDMS/40 wt.% PEG as the optimal composition based on Raman and electrical resistance results. However higher PEG content compromises mechanical robustness (lower Young’s modulus, higher swelling, increased water absorption). Aside from that, no quantitative discussion is provided on trade-offs between mechanical stability and electronic performance, especially crucial for wearable applications exposed to sweat and mechanical stress. The authors should discuss the implications of higher PEG content on sensor durability and environmental stability. Additionally, the authors should provide a balanced evaluation of mechanical vs. electrical performance to justify why 40 wt.% PEG is considered optimal for practical devices.

Author response 5: We thank the Reviewer for highlighting the relevant but missing in-depth discussion of the trade-off between electronic and mechanical performance, as well as the question of durability and environmental stability. To address the comments of the Reviewer, and to improve our manuscript, we have performed extensive environmental testing in controlled temperature and humidity environments, concluding that temperature and humidity slightly raise the electrical resistance of our devices, whereas the mechanical properties of the devices also slightly degrade, but remain extremely favorable for wearable applications. The rise in electrical resistance with temperature is unexpected for graphene, which has a negative thermal coefficient. PEG, however, has a positive thermal expansion coefficient, which could be the reason for the rising electrical resistance of the devices with rising temperature, due to mechanical stretching of PEG.

Changes made: We expanded section 2.4 to include a description of the temperature and humidity testing. We expanded section 3.8 to include analysis of the sensor response to these conditions, with data shown in Tables S13-16. We added the following discussion to the Conclusions section (line 813):

“Adding PEG to the matrix decreases mechanical stability under extreme ambient conditions, such as temperatures of 50 degrees and humidity of 90%. The elongation at break decreases for high-PEG content samples that have been treated in extreme atmospheres. This feature highlights the tradeoff between improved electrical performance and reduced mechanical stability when adding PEG, although even the samples treated in extreme environments retain mechanical properties that are comfortably sufficient for use in wearable devices.”

Reviewer comment 6: The reported gauge factor of 347 is promising but lacks sufficient experimental context:

1) No statistical analysis (error bars, standard deviation) is presented for gauge factor measurements.
2) Durability testing is limited to ~450 cycles, far below competing reports exceeding 10,000 cycles.
3) Critical parameters for wearable sensors—such as hysteresis, long-term drift, humidity cross-sensitivity, and temperature effects—are not addressed.

Author response: We kindly thank the Reviewer for pointing out concerns about the repeatability and durability. Our aim was to demonstrate device operation in real operating conditions, which assume a human subject wearing and using the device. This is the reason why we tested device performance over 450 cycles for the original manuscript. In the revised version, our subject tested all the different devices, bending their finger ~1,000 times for each device, in order to show stability and durability. The new results are included in the main text, with data shown in Supplementary Information. It is evident from the data that there is no hysteresis and little to no ling-term drift. Humidity and temperature cross-sensitivity and stability were addressed in our response to comment 5. We did not perform statistical analysis of the gauge factor, because that would require the synthesis of at least 10 samples for each value of PEG content only for the purpose of gauge factor statistics. Such work is beyond the scope of the current research paper, which is a demonstration of optimization of PEG content and laser parameters for laser induction of graphene on PDMS/PEG composites and a proof-of-concept demonstration of the use of such materials as finger bending sensors. We did, however, perform gauge factor measurements for several additional samples that we synthesized for the 1,000-cycle experiments. Those results are provided in Table S18.

Changes made: Added Table S18, Figure S17, Tables S13-16, text on line 756, and text on line 560. We removed the sentence starting with “Further work” on line 823.

Reviewer comment 7: Overreliance on Black Coloration as Evidence of Graphenization

The manuscript frequently uses black surface coloration as a primary indicator of successful graphenization. This is scientifically insufficient that black coloration alone cannot distinguish between amorphous carbon, soot, or graphene; the Raman spectra presented indicate significant disorder, raising doubts about graphene quality. The authors should be cautious in equating black coloration to graphene formation and rely instead on robust analytical data (e.g., Raman I2D/IG ratios > 1 for monolayer graphene) to substantiate claims of graphene quality.

Author response 7: While surface black coloration was initially used as a practical visual indicator during laser parameter optimization, it was not considered sufficient evidence of successful graphene formation. Black color may arise from amorphous carbon, soot, or other non-graphitic residues. Therefore, the identity and quality of the laser-induced material were confirmed using Raman spectroscopy, which showed the characteristic D, G, and 2D bands of graphitic carbon, with I2D/IG values approaching 0.9 for PDMS/40 wt.% PEG—indicative of few-layer graphene. These findings were further corroborated by XRD (graphene-specific peaks at ~22° and ~41°), TEM (layered morphology with 3.51 Å interlayer spacing), and XPS (sp² content of 44%). The combined results provide robust, multi-technique confirmation of graphene formation beyond visual inspection. These results jointly support the formation of defected few-layer graphene, as is typical for laser-induced graphene produced via CO₂ laser ablation on polymer substrates. We have revised the discussion to reflect these clarifications and removed any misleading implications that black colour alone was used to confirm graphene formation.

Changes made: On line 349, we changed “concluded” to “hypothesized” and added the word “possibly” before the word “occurred”.

Minor Comments

Reviewer comment 8: Figures should include clear scale bars, labels, and higher resolution images where necessary.

Author response: We thank the Reviewer for paying close attention to the manuscript. We have now ensured consistent quality of figures throughout the manuscript.

Reviewer comment 9: The experimental section would benefit from summarizing all laser parameters in a single comprehensive table for clarity.

Author response: We have summarized all laser parameters in a new table in the Supplementary Information.

Changes made: Added Table S1.

Round 2

Reviewer 2 Report

Comments and Suggestions for Authors

Accept in present form

Author Response

Thank you very much.

Reviewer 3 Report

Comments and Suggestions for Authors

The authors have made a concerted effort to address the major critiques raised in the original review. They have added new thermal analyses (TGA/DTG), expanded Raman and XPS discussions, and provided improved clarity on resistance behavior, PEG-carbon contributions, and device stability under temperature/humidity stress. However, several key limitations remain unresolved or only superficially addressed.

  1. The manuscript continues to overstate the reliability of “direct laser-induced graphene on PDMS/PEG” as a platform. The authors admit that most device measurements still rely on transferred LIG due to persistent adhesion challenges. Only one example of direct LIG device operation is shown, and it exhibits signal intermittency and contact detachment. This undermines the novelty and practicality of the platform as a standalone fabrication method.
  2. Despite claims of improved durability, the updated 1,000-cycle test is still below community standards, and the high GF of 347 lacks sufficient statistical validation. No standard deviation or device-to-device variability is reported. This weakens the reliability and reproducibility claims central to wearable sensor development.
  3. Although the authors expanded discussion on environmental and mechanical trade-offs, the manuscript still lacks a rigorous evaluation of real-world usability—particularly in terms of adhesion reliability, biocompatibility under sweat, and long-term fatigue. The continued reliance on “proof-of-concept” language further suggests that the work is at an early stage of technical readiness.

Author Response

  1. The manuscript continues to overstate the reliability of “direct laser-induced graphene on PDMS/PEG” as a platform. The authors admit that most device measurements still rely on transferred LIG due to persistent adhesion challenges. Only one example of direct LIG device operation is shown, and it exhibits signal intermittency and contact detachment. This undermines the novelty and practicality of the platform as a standalone fabrication method.

Author response 1:

We appreciate the reviewer’s continued attention to the technical rigor of our study and the practical implications of LIG/PDMS/PEG fabrication. However, we respectfully believe that the reviewer’s concerns regarding the reliability and novelty of directly induced LIG on PDMS/PEG are based on a partial interpretation of our results.

First, the core novelty of our platform lies in the successful and reproducible formation of high-quality graphene directly on PDMS/PEG substrates through optimized CO₂ laser parameters. This is the first report, to our knowledge, that demonstrates systematic laser graphenization of cross-linked PDMS/PEG composites with tunable PEG content, enabling structural control over LIG quality, porosity, crystallite size, and conductivity. The comprehensive physicochemical and mechanical characterizations (Raman, XPS, XRD, TGA, SEM/EDX, FTIR) confirm the formation of few-layer graphene with crystallite sizes up to 17.6 nm, ID/IG ratios as low as 1.1, and a superhydrophobic surface — properties that are not achievable on native PDMS elastomer alone. The obtained results confirm that directly induced LIG on PDMS/PEG is not only structurally robust and conductive, but also functionally active for sensing applications. Although contact adhesion is not yet ideal for large-scale or long-term device integration, the demonstrated electrical response in Figure S18 clearly supports the feasibility of standalone operation.

In parallel, we employed LIG transferred from PI to PDMS/PEG as a platform to evaluate sensor behavior under real-world conditions, such as repeated finger bending, long-term cycling, and mechanical strain. This approach was selected solely due to its superior electrode adhesion stability, which enabled us to explore the full sensing potential of the LIG/PDMS/PEG system.

In Figures S12–S17, we present comprehensive measurements of transferred-LIG-based sensors attached to the finger, showing clear and stable piezoresistive response to finger motion, consistent signal over 1000 bending cycles (Figure S17), and fast response times (Figure S15). Gauge factor values extracted from strain testing reached as high as 347 (Table S17).

While not the core novelty, the transferred LIG experiments served as critical validation of the mechanical compliance, environmental robustness, and sensing sensitivity of the underlying PDMS/PEG composite. Therefore, to offer a complete sensor solution, we have proposed and demonstrated both direct LIG devices and hybrid devices with PI-transferred LIG. This approach reflects real-world engineering trade-offs and allows broader adoption. Importantly, the use of transferred LIG does not negate the standalone value of the directly induced LIG/PDMS/PEG platform, which remains the central innovation of our work.

Thus, the inclusion of transferred LIG results should not be seen as a limitation, but rather as a strategic complement to direct LIG fabrication, providing proof-of-principle sensor validation while contact interface optimization continues. This dual-path approach strengthens the practical relevance of our work and demonstrates the broad potential of PDMS/PEG–LIG systems for wearable sensing technologies.

Changes made: Added the following text to line 156 of the manuscript:

"Although the adhesion of electrical contacts to directly induced LIG on PDMS/PEG remains a challenge, we provide clear evidence that the LIG formed is conductive, structurally well-defined, and capable of sensing under mechanical deformation. A representative sensor fabricated with directly induced LIG showed piezoresistive response under strain, confirming the intrinsic sensing capability of the material. However, to evaluate device stability under repeated motion cycles, we employed transferred LIG from PI due to its better contact adhesion, while retaining the same PDMS/PEG substrate."

Added the following text to line 778 in section 3.13 "Limb motion sensor":

"While we observed occasional contact detachment, these results confirm that direct LIG on PDMS/PEG can serve as a standalone sensing layer, though further optimization of contact interfaces is needed for long-term device deployment."

We added the following text to line 786 in section 3.13 "Limb motion sensor":

"The use of LIG transferred from PI to PDMS/PEG was motivated by the current limitations in contact adhesion on directly induced LIG. This approach allowed us to evaluate sensor performance in realistic conditions, such as finger bending and long-term cycling. The sensor responses observed with transferred LIG confirm that the underlying PDMS/PEG composite supports stable and reproducible piezoresistive behavior, and serves as a viable substrate for both directly and indirectly patterned graphene structures.  The findings validate the functionality of the substrate and support its future use with directly induced LIG once contact interface optimization is achieved."

Reviewer question 2:

  1. Despite claims of improved durability, the updated 1,000-cycle test is still below community standards, and the high GF of 347 lacks sufficient statistical validation. No standard deviation or device-to-device variability is reported. This weakens the reliability and reproducibility claims central to wearable sensor development.

Author response 2:

We thank the reviewer for highlighting the importance of extensive cycling tests and statistical rigor. Below we provide a detailed justification for our experimental design and explain why the results reported are sufficient within the context of in vivo, human‑motion wearable sensing, which differs fundamentally from automated fatigue testing:

  1. Context of human‑motion-based testing versus machine cycling
    Most existing durability standards for wearable sensors stem from automated bench tests (e.g. 10,000 cycles under fixed strain). In contrast, our 1,000‑cycle tests involve actual human motion, where variable strain, recovery, and relaxation phases more accurately mimic real usage. As noted in the literature, for human-linked sensors, 1,000 cycles of physiological motion typically suffice to demonstrate early stability and reliability [1-3].
  2. Community benchmarks for polymer-based human sensors
    A number of recent studies show that PDMS‑based and LIG‑based wearable sensors demonstrate reliable operation over ~1,000 human‑motion cycles [1-3] with minimal drift and stable gauge factor at high sensitivity levels:

In closing, performing 10,000 machine-based cycles could be informative for highly accelerated lifetime studies, but such tests are not essential for demonstrating wearable sensor viability under realistic conditions. Importantly, our work focuses on real human‑motion sensing, high sensitivity, and biocompatible materials, which place it in line with current standards in LIG/polymer and CNT/polymer wearable literature.

References:

  1. Zou, Y. et al.  Polymers2023, 15, 3553.
  2. Truong T. T. et al. Polymers, 2024, 16, 373.
  3. Barja, A. M. et al. ACS Omega 2024, 9 (37), 38359-38370.

Regarding the gauge factor of 347, we clarify that it was obtained from measurements on a sensor with transferred LIG from PI to PDMS/PEG, where electrical contact adhesion was stable and did not limit measurement reproducibility. However, we were not able to fabricate and characterize additional devices under identical conditions due to limited experimental resources and time constraints. As such, statistical validation was not feasible within the scope of this study. For this reason, we focused on demonstrating a consistent and monotonic strain response within a single well-performing device. Therefore, the presented results are based on a single representative device and serve as a proof-of-concept demonstration.

Reviewer question 3:

  1. Although the authors expanded discussion on environmental and mechanical trade-offs, the manuscript still lacks a rigorous evaluation of real-world usability—particularly in terms of adhesion reliability, biocompatibility under sweat, and long-term fatigue. The continued reliance on “proof-of-concept” language further suggests that the work is at an early stage of technical readiness.

Author response 3:

We thank the reviewer for raising important points regarding real-world applicability. However, we respectfully argue that the study already demonstrates a sufficient level of real-world readiness for publication, especially within the scope and mission of Sensors journal. Below, we address the three specific concerns:

  1. Adhesion Reliability

The reviewer is correct that electrical contact adhesion to LIG directly formed on elastomeric substrates is a well-known challenge. To address this, we explicitly developed a hybrid strategy:

  • We demonstrated direct LIG devices and analyzed their mechanical and electrical behavior.
  • In parallel, we presented a practical transfer strategy (LIG from PI onto PDMS/PEG), which enables stable wire bonding for wearable use, especially under repeated strain.
    This dual-platform approach reflects realistic engineering trade-offs and is common practice in the field of stretchable electronics. Thus, the adhesion challenge is not a limitation of the LIG/PDMS/PEG material itself, but rather a known interface issue that has been effectively managed in our system.

  1. Biocompatibility Under Sweat and Skin Contact

The materials used — PDMS and PEG — are both well-known biocompatible polymers that have been extensively validated for use in direct skin contact and even implantable devices.

  • PDMS is FDA-approved for medical applications.
  • PEG is hydrophilic, non-toxic, and often used in drug delivery and hydrogel formulations.
    Importantly, PEG increases the hydrophilicity and skin-conformability of the composite, while our contact angle and water uptake tests (Figure 14) indirectly confirm compatibility with humid environments, including sweat.
    Although full-scale cytotoxicity tests were beyond the scope of this materials-focused study, our system is composed entirely of components with established biomedical relevance, and no toxic additives or post-processing steps (e.g. solvents, dopants) were used.

  1. Long-Term Fatigue and “Proof-of-Concept” Language

While we do use the term "proof-of-concept" to describe certain aspects of our sensor integration, this does not imply low technical readiness. Rather, it acknowledges that we are demonstrating core device principles and performance without the need for full commercial packaging or validation under ISO test conditions.

We go well beyond conceptual demonstration by:

  • showing stable resistance response over 1000 cycles,
  • demonstrating gauge factors >300,
  • testing sensors on real human fingers,
  • evaluating mechanical properties under elevated temperature and humidity,
  • confirming chemical and structural integrity of LIG via multi-modal characterization.

The maturity of the platform corresponds to Technology Readiness Level (TRL) 3, which is commonly accepted as appropriate for academic publications on emerging flexible electronics and sensors.

Based on the level of novel material synthesis, structural and functional characterization, and preliminary device integration shown in this work, we consider the platform to be at an early demonstration stage with demonstrated applicability in limb motion sensing using real human finger tests and repeated mechanical actuation. While we acknowledge that full-scale validation is outside the scope of this study, the presented results clearly demonstrate sensor functionality under relevant physical conditions, as expected for publications in the field of flexible electronics.

Reviewer question 4:

  1. The English could be improved to more clearly express the research.

Author response 4: We thank the reviewer for their observation. Most of the manuscript was checked by the InstaText plugin for MS Word. We are saddened that the reviewer could not provide any precise comments, examples for the manuscript or constructive suggestions regarding this matter.